# Sensitivity of yeast to lithium chloride connects the activity of *YTA6* and *YPR096C* to translation of structured mRNAs

**Maryam Hajikarimlou[1], Houman Moteshareie[1], Katayoun Omidi[1], Mohsen Hooshyar[1], Sarah Shaikho[2], Tom Kazmirchuk[1], Daniel Burnside[1], Sarah Takallou[1], Narges Zare[1], Sasi Kumar Jagadeesan[1], Nathalie Puchacz[1], Mohan Babu[3], Myron Smith[1], Martin Holcik[4], Bahram Samanfar[1,5], Ashkan Golshani[1]***

**1** Department of Biology and Ottawa Institute of Systems Biology, Carleton University, Ottawa, Ontario, Canada, **2** Molecular Biomedicine Program, Children's Hospital of Eastern Ontario Research Institute, Ottawa, Ontario, Canada, **3** Department of Biochemistry, Research and Innovation Centre, University of Regina, Regina, Canada, **4** Department of Health Sciences, Carleton University, Ottawa, Ontario, Canada, **5** Agriculture and Agri-Food Canada, Ottawa Research and Development Centre (ORDC), Ottawa, Ontario, Canada

* Ashkan.Golshani@carleton.ca

**Data Availability Statement:** All relevant data are within the manuscript and its Supporting Information files.

## Abstract

Lithium Chloride (LiCl) toxicity, mode of action and cellular responses have been the subject of active investigations over the past decades. In yeast, LiCl treatment is reported to reduce the activity and alters the expression of *PGM2*, a gene that encodes a phosphoglucomutase involved in sugar metabolism. Reduced activity of phosphoglucomutase in the presence of galactose causes an accumulation of intermediate metabolites of galactose metabolism leading to a number of phenotypes including growth defect. In the current study, we identify two understudied yeast genes, *YTA6* and *YPR096C* that when deleted, cell sensitivity to LiCl is increased when galactose is used as a carbon source. The 5'-UTR of *PGM2* mRNA is structured. Using this region, we show that *YTA6* and *YPR096C* influence the translation of *PGM2* mRNA.

## Introduction

Dysregulation of signaling pathways in the brain is thought to be the main cause of bipolar disorder (BD) [1]. Lithium chloride (LiCl) has remained an important treatment option for BD for decades [2,3]. It has been prescribed to prevent both new depressive and manic episodes and is known to be the only compound to have anti-suicidal effects in BD patients [4].

When LiCl is used as a therapeutic agent, it is generally accepted that in the short term, it influences Protein Kinase C (PKC) and glycogen synthesis kinase-3 (GSK-3) signal transduction pathways. Long term exposure to LiCl modifies the expression of different genes/pathways including PI/PKC signaling cascade, leading to alterations in the synaptic function of the nerve cells [1,5–7]. Inducing autophagy, oxidative metabolism, apoptosis and affecting translation machinery are other pathways proposed to be influenced by LiCl intake [2,6]. LiCl has

**Funding:** This research was funded by Natural Sciences and Engineering Research Council of Canada, NSERC grant number: 123456.

**Competing interests:** The authors have declared that no competing interests exist.

also been investigated as a treatment option for Alzheimer's disease which is caused by the aging of the nervous system [6,8]. Although much has been learned about the influence of LiCl, how it affects the cell at the molecular level and the mechanism(s) of its activity, as well as its side effects (secondary effects) require further investigations [1,2,8].

At the molecular level, the sensitivity of yeast cells to LiCl was previously described by changes in the level of expression and activity for *PGM2* that encodes a phosphoglucomutase [9,10]. Phosphoglucomutase is responsible for converting glucose-1-phosphate to glucose-6-phosphate and LiCl is an inhibitor of its enzymatic activity. When galactose is used as the carbon source, inhibition of phosphoglucomutase by LiCl results in the accumulation of galactose metabolite intermediates that in turn causes growth defects [11,12]. In the presence of glucose, LiCl reduces the levels of UDP-glucose and disrupts the associated pathways. It has also been suggested that LiCl may inhibit RNA processing enzymes [13,14]. Also, it is reported that under LiCl stress, there seems to be a rapid loss of ribosomal protein gene pre-mRNAs and a decrease in the number of mature mRNAs in the cytoplasm [14]. In addition, it is possible that LiCl may inhibit the initial steps of the protein synthesis pathway. It is thought that LiCl may disrupt the association of translation initiation factor eIF4A RNA helicase to the yeast translation machinery [9] impairing translation initiation. Deletion of *TIF2* that codes for the eIF4A helicase increased yeast sensitivity to LiCl. Over-expression of eIF4A helicase reverted the translational inhibition caused by LiCl [9].

In the current study, we observed that the deletion of two yeast genes, *YTA6* and *YPR096C* increased the sensitivity of yeast cells to LiCl. *YTA6* codes for a putative ATPase of the CDC48/PAS1/SEC18 (AAA) family of proteins and *YPR096C* codes for a protein of unknown function. Neither of the genes was previously linked to cell responses to LiCl. Our follow-up genetic investigations suggest that the involvement of *YTA6* and *YPR096C* in yeast LiCl sensitivity seems to be due to their influence on *PGM2* translation.

## Materials and methods

### Strains, plasmids, gene collections and cell and DNA manipulations

MATa mating strain Y4741 orfΔ::KanMAX4 his3Δ1 leu2Δ0 met15Δ0 ura3Δ0 and MATα mating strain, Y7092 can1Δ::STE2pr-Sp_his5 lyp1Δ his3Δ1 leu21Δ0 ura3Δ0 met15Δ0 were used. Yeast non-essential gene knockout collections [15], yeast over-expression plasmid library [16] and the collection of yeast gene-GFP fusion strains were utilized as before [17–19]. Yeast gene knockout was performed by PCR transformation using the Lithium Acetate method and confirmed by PCR analysis [20,21]. Over-expression plasmids for *YTA6* and *YPR096C* were purchased from Thermofisher® and their integrity was confirmed using PCR analysis. *PGM2*-GFP strain was purchased from Thermofisher® Yeast GFP Clone Collection and was utilized in qRT-PCR and western blot analysis. The integrity of this strain was confirmed using PCR and drug sensitivity analyses.

p281 construct carries a *LacZ* expression cassette under the control of a gal promoter. p281-4 construct carries an insert with a strong hairpin structure (5′ GATCCTAGGATCCTAG–GATCCTAGG ATCCTAG3′) upstream of *LacZ* cassette[22]. pAG25 plasmid was used as a DNA template for nourseothricin sulfate (clonNAT) resistance gene marker in PCR reactions for gene knockout experiments. Kanamycin and NAT markers were used as selection markers for corresponding deletion mutant strains. All plasmids carried an ampicillin resistance gene which was used as a selection marker in *E. coli* DH5α, and a *URA3* marker gene for selection in yeast.

P416 construct carries a *LacZ* expression cassette under the transcriptional control of a gal promoter. To generate reporter *LacZ* mRNAs under the translational control of complex RNA

structures, three different fragments were cloned upstream of the *LacZ* mRNA in p416 construct using *XbaI* restriction site. In this way three expression constructs were designed as follows: pPGM2 construct contains the 5'-UTR of *PGM2* gene (5' TAATAAGAAAAAGATCAC CAATC TTTCTCAGTAAAAAAAGAACAAAAGTTAACATAACAT 3'), pTAR construct contains the 5' UTR of *HIV1-tar* gene (5' GGGTTCTCTGGTTAGCCAGATCTGAGCCCGGGAGC TCTCTGGCTAGCTAGGGAACC CACTGCTTAAGCCTCAATAAAGCTTGCCTTGAGTGCTTCA AGTAGTGTGTGCC 3') and pRTN that contains the 5' UTR of *FOAP-11* gene (5' GGGATTT TTACATCGTCTTGGTAAAGGCGTGTGACCCATA GGTTTTTTAGATCAAACACGTCTTTACA AAGGTGATCTAAGTATCTC 3').

YP (1% Yeast extract, 2% Peptone) or SC (Synthetic Complete) with selective amino acids (0.67% Yeast nitrogen base w/o amino acids, 0.2% Dropout mix,) either with 2% dextrose or 2% galactose, as a source of carbohydrates, was used as culture medium for yeast and LB (Lysogeny Broth) was used for *E. coli* cultures. 2% agar was used for all solid media. Yeast plasmid extraction was performed using yeast plasmid miniprep kit (Omega Biotek®) and *E. coli* plasmid extraction was carried out using GeneJET plasmid miniprep kit (Thermofisher® and Bio-Basics®) according to the manufacturers' instructions.

### Drug sensitivity analysis

For drug sensitivity analysis, yeast cells were grown from independent colonies to saturation for two days at 30˚C in liquid YPgal. Spot test analysis of serial dilutions of cell suspensions were spotted onto solid media with or without LiCl. For growth sensitivity to LiCl, 10 mM and 100 mM concentrations were used in media containing galactose or glucose, respectively, as described before [10,11]. Sensitivity to the compound was assessed by comparing the number and size of the colonies formed on each plate after 48 hours in comparison with wild type [20].

For quantification analysis, colony counting was done by taking 100 μL of diluted ($10^{-4}$) cell cultures from independent colonies, grown for two days at 30˚C in liquid YPgal, and spreading on YPgal plates in the absence and presence of LiCl. The colonies were counted two days after incubation at 30˚C. Each experiment was repeated at least three times. t-test analysis ($P$-value ≤ 0.05) was used to determine statistically significant differences.

### Quantitative *β-galactosidase* assay

The effect of 5'-UTR regions to mediate translation in different yeast strains were examined using *LacZ* reporter systems. To evaluate the activity of *LacZ* expression cassettes, quantitative *β-galactosidase* assay was performed using ONPG (O-nitrophenyl-α-D-galactopyranoside) as described [23,24]. Each experiment was repeated at least three times.

### Quantitative real time PCR (qRT-PCR)

The content of mRNAs was evaluated using qRT-PCR analysis. Deletion mutants in *PGM2*-GFP strain background were grown in YPgal overnight with or without LiCl treatment. Total RNA was extracted with Qiagen® RNeasy Mini Kit. Complementary DNA (cDNA) was made using iScript Select cDNA Synthesis Kit (Bio-Rad®) according to the manufacturer's instructions. cDNA was then used as a template for quantitative PCR. qPCR was carried out using Bio-Rad® iQ SYBR Green Supermix and the CFX connect real time system (Bio-Rad®), according to the manufacturer's instructions. *PGK1* was used as a constitutive housekeeping gene (internal control). The procedure and data analysis were performed according to MIQE guidelines [25].

The procedure was done in three repeats and t-test analysis (*P*-value $\leq$ 0.05) was used to determine statistically significant results. The following primers were used to quantify *PGM2* and *PGK1* mRNAs, as our positive control in different mutant strains.

*PGM2*: Forward `GGTGACTCCGTCGCAATTAT`; Reverse: `CGTCGAACAAAGCACAGAAA`
*PGK1*: Forward `ATGTCTTTATCTTCAAAGTT`; Revers: `TTATTTCTTTTCGGATAAGA`

## Western blot analysis

Western blot analysis was used to investigate the protein content for Pgm2p-GFP fusion protein. Different strains were grown in media treated with and without LiCl. Protein extraction was performed as described by Szymanski [26]. Bicinchoninic acid assay (BCA) was performed to estimate protein concentration as described by the manufacturer (Thermo Fisher®). Equal amounts of total protein extract (50 µg) were loaded onto a 10% SDS-PAGE gel, run on Mini-PROTEAN Tetra cell electrophoresis apparatus system (Bio-Rad®). Proteins were transferred to a nitrocellulose 0.45 µm membrane via a Trans-Blot Semi-Dry Transfer (Bio-Rad®). Mouse monoclonal anti-GFP antibody (Santa Cruz®) was used to detect protein levels of Pgm2p-GFP. Mouse anti-Pgk1 (Santa Cruz®) was used to detect Pgk1 protein levels used as internal controls. Immunoblots were visualized with chemiluminescent substrates (Bio-Rad®) on a Vilber Lourmat gel doc Fusion FX5-XT (Vilber®). Densitometry analysis was carried out using the FUSION FX software (Vilber®). Experiments were repeated at least three times; t-test analysis (*P*-value $\leq$ 0.05) was used to determine statistically significant results.

## Genetic interaction analysis

Synthetic genetic analysis for *YTA6* and *YPR096C* was performed in a 384 format as before [17,19,27]. In brief, deletion mutant for query genes in Mat α mating type were crossed to two sets of gene deletion mutants in Mat a mating type. After a few rounds of selection, double gene deletion mutants were selected in Mat a mating type. Colony size was used as a measure of fitness [27,28]. Colony size was measured as described before [29,30]. The experiment was repeated three times.

For Phenotypic Suppression Array (PSA) analysis a MATα yeast strain having an over-expression plasmid of our query gene is mated into the entire deletion set along with an empty plasmid used as a control [31,32]. For phenotypic suppression analysis, the final constructs transformed into deletion library were grown on YPgal compared to the control plasmid. Phenotypic suppression array was performed by growing the transformed cells on YPgal with a sub-inhibitory concentration of LiCl (3 mM, approximately 1/3 of the concentration used for strain sensitivity analysis) as a stress condition drug [33]. We investigated the ability of the over-expression of our query genes to compensate for the sick phenotype of our deletion sets under the inhibitory concentration of LiCl. If the over-expression of our candidate genes overcome the sensitivity of a yeast deletion strain caused by drug inhibition, we can suggest that a functional connection exists between the two genes [20,34].

## Genetic interaction data analysis

Scoring fitness was done by colony size measurement as in [29,30]. Those deletions that had 30% or more reduction in colony size in at least two experiments were considered hits. Based on their biological process and/or molecular function, hits were clustered into groups with enriched GO terms using Gene Ontology Resource http://geneontology.org/ and Genemania database http://genemania.org.

## Results and discussion

### Deletion of *YTA6* and *YPR096C* increases yeast sensitivity to lithium

Drug sensitivity of mutant strains to a target chemical is an important tool to investigate how a chemical compound affects the cell at the molecular level and pathways influenced by the drug [17,19,35]. While investigating yeast gene deletion mutants that are sensitive to LiCl we identified two gene deletion mutants for *YTA6* and *YPR096C* that showed increased sensitivity to LiCl. Little is known about the molecular activity of these two genes and the cellular process in which they participate making them interesting targets to study. *YTA6* codes for a putative ATPase and *YPR096C* is an uncharacterized ORF.

In the spot test assay indicated in Fig 1 *yta6Δ* and *ypr096cΔ* show growth reduction in the presence of LiCl (10 mM LiCl) suggesting increased sensitivity of yeast strains when these two genes are deleted. *tif2Δ* was used as a positive control. Introduction of the over-expression plasmids that express the deleted genes, into the corresponding gene deletion mutants reversed the observed sensitivities to LiCl (Fig 1). To confirm the results obtained by the spot test assay we perform colony count measurement analysis, which represents a more quantitative approach. In this method, the decreased percentage of colonies is calculated by dividing the number of colonies in media in the presence of the LiCl to the number of colonies in control media and normalized to Wild Type (WT). Indicated in Fig 2 deletion of *YTA6*, *YPR096C* or *TIF2* show reduced colony formation in the presence of LiCl. As before, introduction of the over-expression plasmids that express the deleted genes into the corresponding gene deletion mutants suppressed cell sensitivities to LiCl caused by gene deletions.

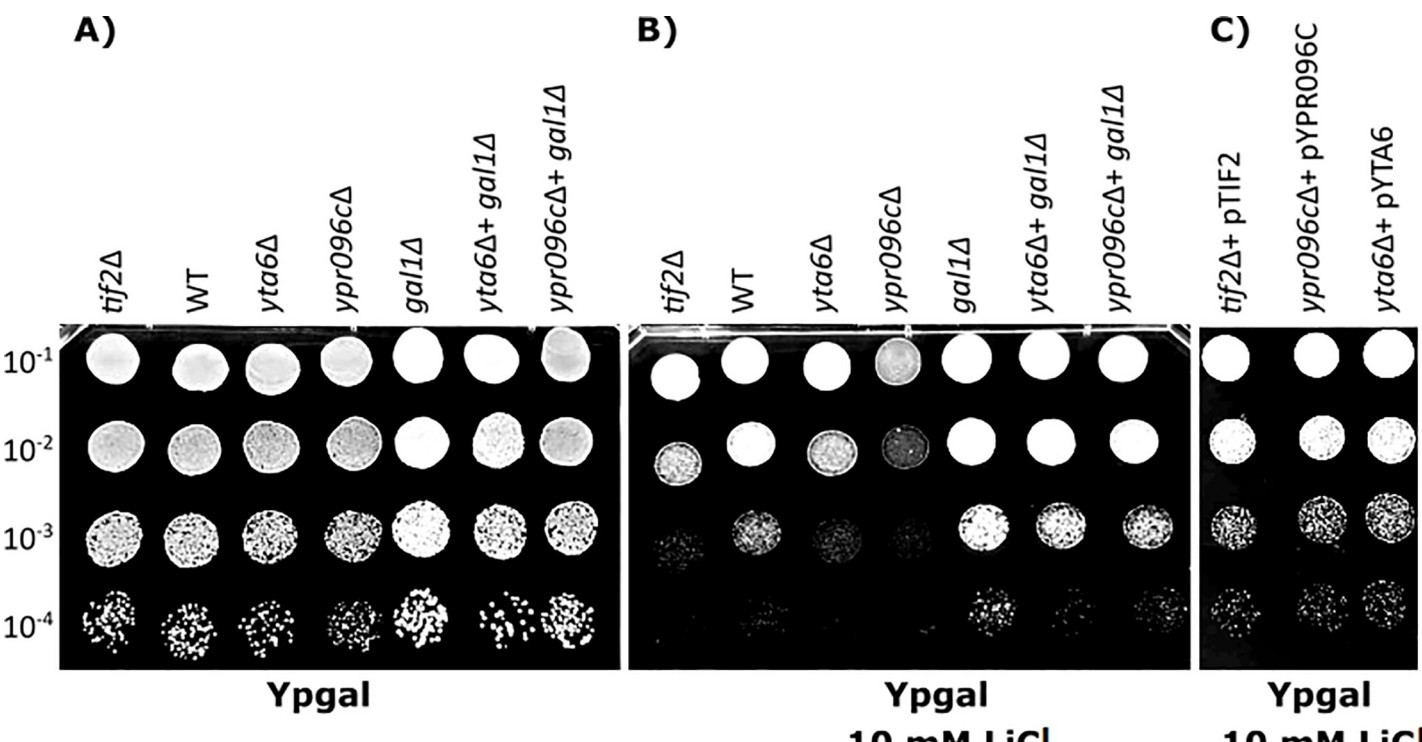

**Fig 1. Drug sensitivity analysis for different yeast strains using spot test assay.** In (A) and (B) yeast cells were serially diluted as indicated ($10^{-1}$ to $10^{-4}$) and spotted on YPgal media with or without LiCl (10 mM). *yta6* and *ypr096c* show less growth under LiCl treatment. Double deletion for *GAL1* with *YTA6* or *YPR096C* suppressed the observed sensitivity of single-gene deletions for *YTA6* or *YPR096C*. Deletion of *TIF2* was used as a positive control. In (C) over-expression of the target gene in their corresponding deletion mutants reverted cell sensitivity to LiCl (10 mM). Each experiment was repeated at least three times (n $\geq$ 3) with similar outcomes.

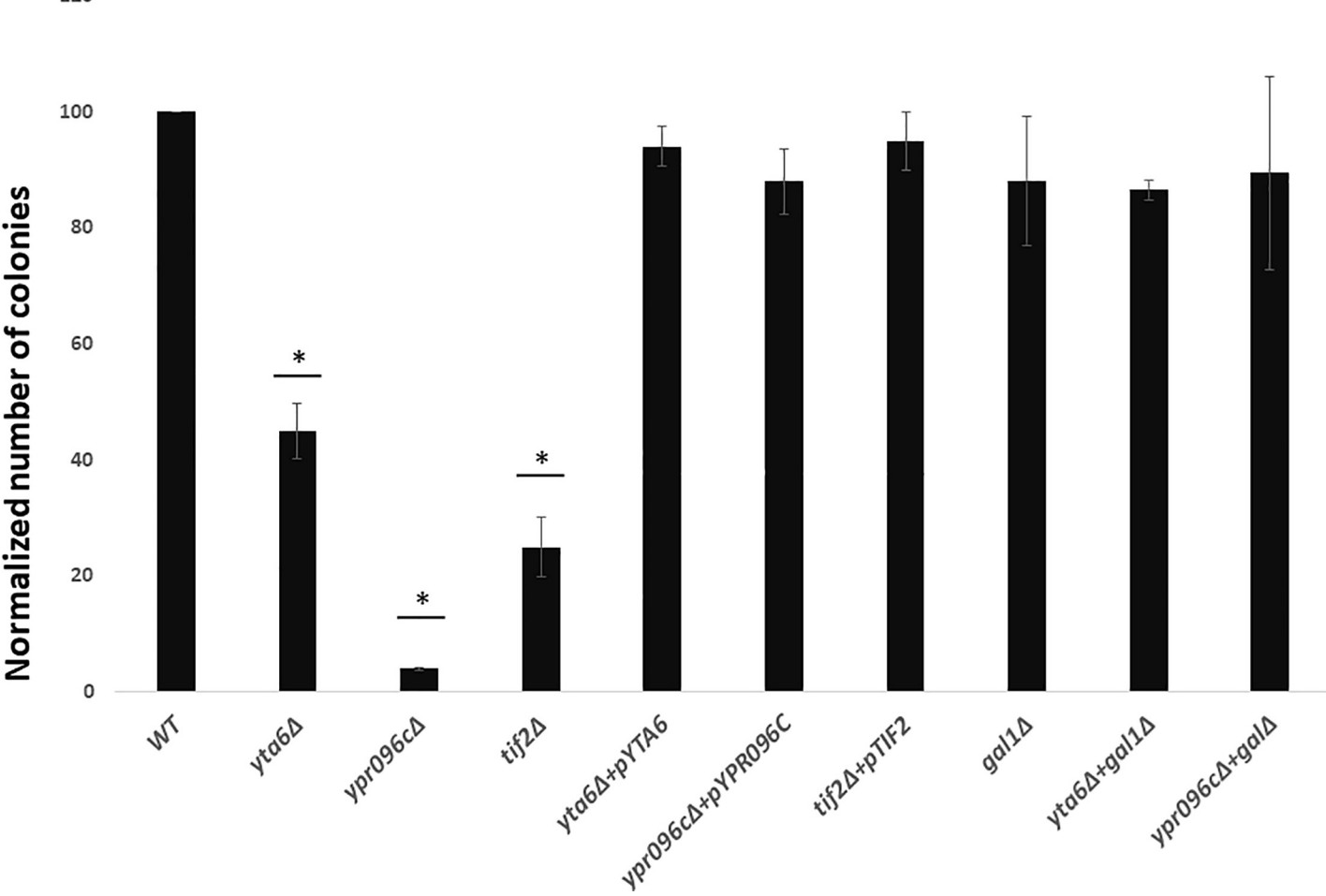

**Fig 2. Quantitative analysis of drug sensitivity for different yeast strains.** The average number of colonies formed for different yeast strains in the presence of LiCl (10 mM) was normalized to that for the WT strain (WT average colony count = 285.33). Double deletion for *GAL1* with *YTA6* or *YPR096C* suppressed the observed sensitivity of single-gene deletions for *YTA6* or *YPR096C*. Data represent the average from three independent experiments (n = 3) and error bars represent standard deviation. * represent statistically significant results compared to the WT. t-test analysis (*P*-value ≤ 0.05) was used to compare differences.

LiCl reduces the activity of phosphoglucomutase enzyme leading to the accumulation of intermediate metabolites from the galactose metabolism including galactose-1-phosphate, a toxic intermediate. In yeast, galactokinase is encoded by the *GAL1* gene. To investigate the influence of *YTA6* and *YPR096C* on LiCl toxicity through galactose metabolism, we generated double gene deletions for *YTA6* or *YPR096C* with the *GAL1* gene. Deletion of the *GAL1* gene relieved the sensitivity of gene deletion mutants for *YTA6* or *YPR096C* to LiCl (Fig 1). Also, when glucose was used as a carbon source deletion strains for *YTA6* or *YPR096C* showed no sensitivity to 10 mM LiCl. When the concentration of LiCl was increased to a toxic level (100 mM) in the presence of glucose as a carbon source [11,36], deletion mutants for *YTA6* or *YPR096C* did not show increased sensitivity (S1 Fig). Together these results further connect the observed LiCl sensitivity for *YTA6* and *YPR096C* deletion strains to galactose metabolism.

### *YTA6* and *YPR096C* regulate the expression of *PGM2* at the level of translation

*PGM2* has been identified as a target of LiCl in yeast cells and its expression has been reported to change in the presence of LiCl [10]. Next, we investigated the ability of *YTA6* and *YPR096C*

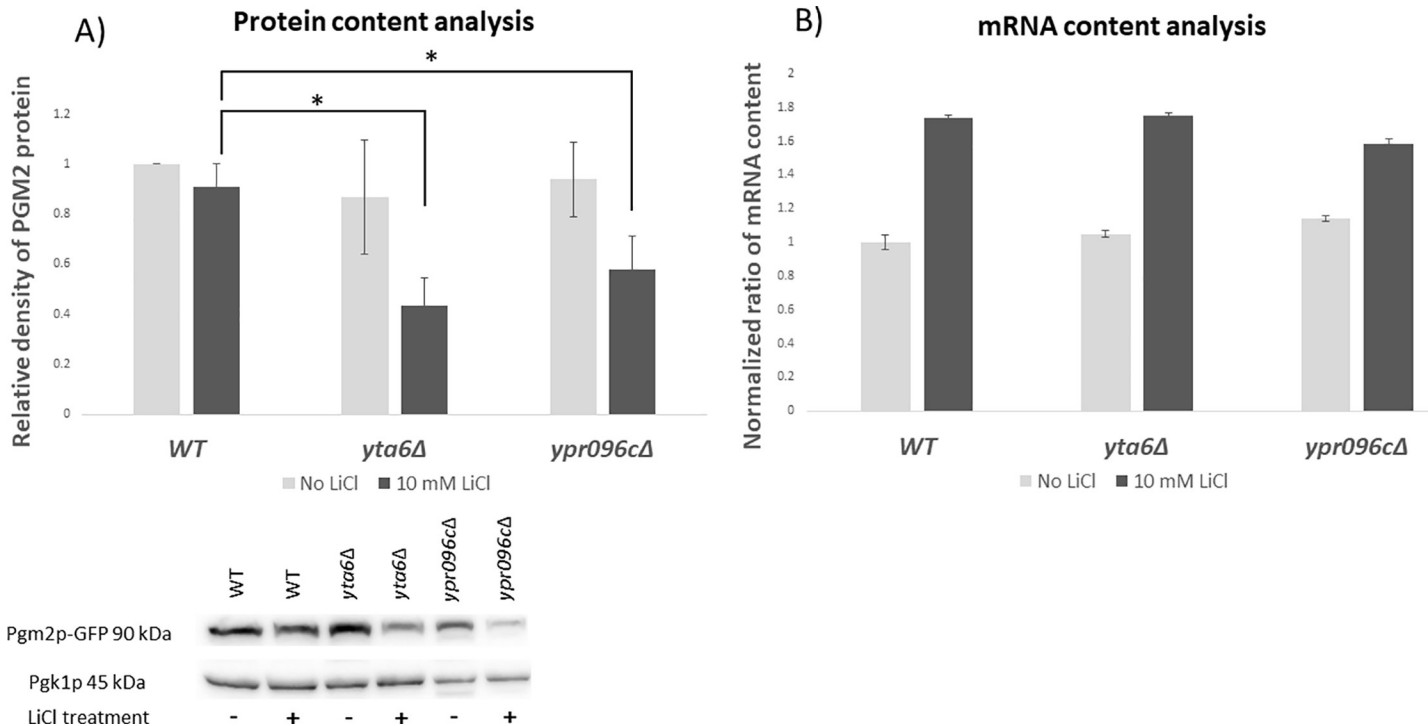

**Fig 3. Protein and mRNA content analysis.** (A) Protein content analysis of Pgm2p-GFP protein in deletion of yeast strains for *yta6Δ* and *ypr096cΔ*. Western blot analysis was used to measure the protein content for Pgm2p-GFP protein in the absence or presence of LiCl (10 mM) and related to WT. Pgk1p was used as a housekeeping gene and the values are normalized to that. The inset represents a typical blot (B) mRNA content analysis of *PGM2* in *yta6Δ* and *ypr096cΔ*. qRT-PCR was used to evaluate the content of *PGM2* mRNA in yeast gene deletion mutants related to WT strain and normalized to *PGK1* mRNA levels in the absence or presence of LiCl (10 mM). Each experiment was repeated at least three times (n ≥ 3). Error bars represent standard deviation. * represent statistically significant results compared to the value in the corresponding WT. t-test analysis (*P*-value ≤ 0.05) was used to compare differences.

to change *PGM2* expression both at the levels of translation (Fig 3A) and transcription (Fig 3B). This was done using western blot analysis in a strain where Pgm2p was tagged with a *GFP* gene. In the absence of LiCl, we observed no notable alteration in the Pgm2p levels when either *YTA6* or *YPR096C* were deleted. However, when cells were challenged with 10 mM LiCl, the deletion of either *YTA6* or *YPR096C* reduced the protein content of Pgm2p.

To investigate the possible effect of *YTA6* and *YPR096C* on *PGM2* transcription, we used qRT-PCR analysis to measure the content of *PGM2* mRNA when *YTA6* and *YPR096C* were deleted. Indicated in Fig 3B, deletion of *YTA6* and *YPR096C* did not noticeably change the content of *PGM2* mRNA when cells were treated with LiCl. This suggests that *YTA6* and *YPR096C* are unlikely to alter *PGM2* expression at the transcription level. Together these observations connect the activities of *YTA6* and *YPR096C* to the expression of Pgm2p at the protein level. This is in agreement with a previous observation by Sofola-Adesakin et al. that in *Drosophila melanogaster* LiCl impaired gene expression at the protein synthesis level and not the mRNA level[6].

## Translation of *β-galactosidase* reporter mRNA with a hairpin structure is altered by the deletion of *YTA6* and *YPR096C*

The 5'-UTR of *PGM2* mRNA is predicted to contain a highly structured region [37,38] (S2 Fig). This knowledge along with the observation that *YTA6* and *YPR096C* appear to impact *PGM2* expression at the translation level prompted us to investigate the influence of *YTA6* and

*YPR096C* on the translation of other structured mRNAs. First, we placed the 5'-UTR of *PGM2* mRNA in front of a *LacZ* reporter gene in a p416 expression construct [39]. Indicated in Fig (4A and 4B), when *YTA6* and *YPR096C* were deleted the activity of *β-galactosidase* was reduced for the reporter gene that contained 5'-UTR of *PGM2* mRNA and not a control mRNA without the 5'-UTR of *PGM2*. The deletion of *TIF2* was used as a positive control.

Next, we utilized an expression cassette, p281-4 with a strong hairpin structure in front of a *LacZ* reporter gene [22]. A second construct, p281 without the hairpin structure was used as a control. Illustrated in Fig (5A and 5B) it was observed that when *YTA6* and *YPR096C* were deleted the activity of *β-galactosidase* was reduced for the reporter gene that contained a hairpin structure. When the hairpin was absent, the activity of *β-galactosidase* was independent of *YTA6* and *YPR096C*. Together these data show that the deletion of *YTA6* and *YPR096C* seem to reduce the translation of structured reporter mRNAs.

Next, we investigated the influence of *YTA6* and *YPR096C* on other structured mRNAs. For this, we designed two additional *β-galactosidase* mRNA reporters each carrying different complex RNA structures. pTAR carries the 5'-UTR of the *HIV1-tar* gene. This region contains a strong hairpin loop involved in modulating expression [40]. pRTN carries the 5' UTR of *FOAP-11* gene that contains a highly structured region [41]. Indicated in (Fig 5C and 5D), deletion strains for *YTA6* and *YPR096C* had a reduced level of *β-galactosidase* expression.

## Genetic interaction analysis further connects the activity of *YTA6* and *YPR096C* to the protein biosynthesis pathway

Genetic Interaction (GI) analysis is based on the assumption that parallel compensating cellular pathways give the cell its plasticity and tolerance against random deleterious mutations [29]. In this way, deletion of individual genes that can functionally compensate for each other has little or no phenotypic consequences. However, when both genes are deleted, an unexpected phenotype can emerge which can often be detected but a decrease in cell fitness or even cell death. In this case, the two genes are said to be forming a negative genetic interaction (nGI). An nGI can reveal the involvement of genes in compensating parallel pathways. nGI analysis has been used in various investigations to study gene function [17,18,33]. Systematic analysis of GIs in yeast is made possible by its two mating types. A target gene deletion in α-mating type (MAT alpha) is crossed with an array of single-gene deletion in a-mating type (MAT a) background and after a few rounds of selection double gene knockouts are selected [27]. Colony size measurement is often used to determine the fitness of double gene knockouts [28]. To this end, we generated a set of double gene deletions mutants for our two query genes with 402 deletion mutants for genes involved in gene expression (S2 Table). This array was termed gene expression array. Due to inherent bias associated with such enriched subsets, a second set of double gene deletions were made for our query genes with 304 random gene deletions, termed random array, and was used as a control (S2 Table).

*YTA6* formed 7 nGIs with different genes (S3 Table). The list of interactors includes *YPL079W* that encodes for large ribosomal subunit protein 21B and *YPL090C* that codes for small ribosomal subunit 6A. *YPR096C* interacted with 8 genes including *YOR091W* that codes for a protein associated with translating ribosomes and *YOR078W* that codes for a protein involved in small ribosomal subunit biogenesis (S3 Table). The low number of nGIs observed for both *YTA6* and *YPR096C* makes it difficult to draw a statistically meaningful enrichment for the interacting genes. As a result, formulating function(s) for *YTA6* and *YPR096C* on the basis of the observed interactions is not feasible.

In addition, we also investigated the conditional nGIs for the two target genes. Conditional GIs represent an interesting form of gene association. They represent a further insight into the

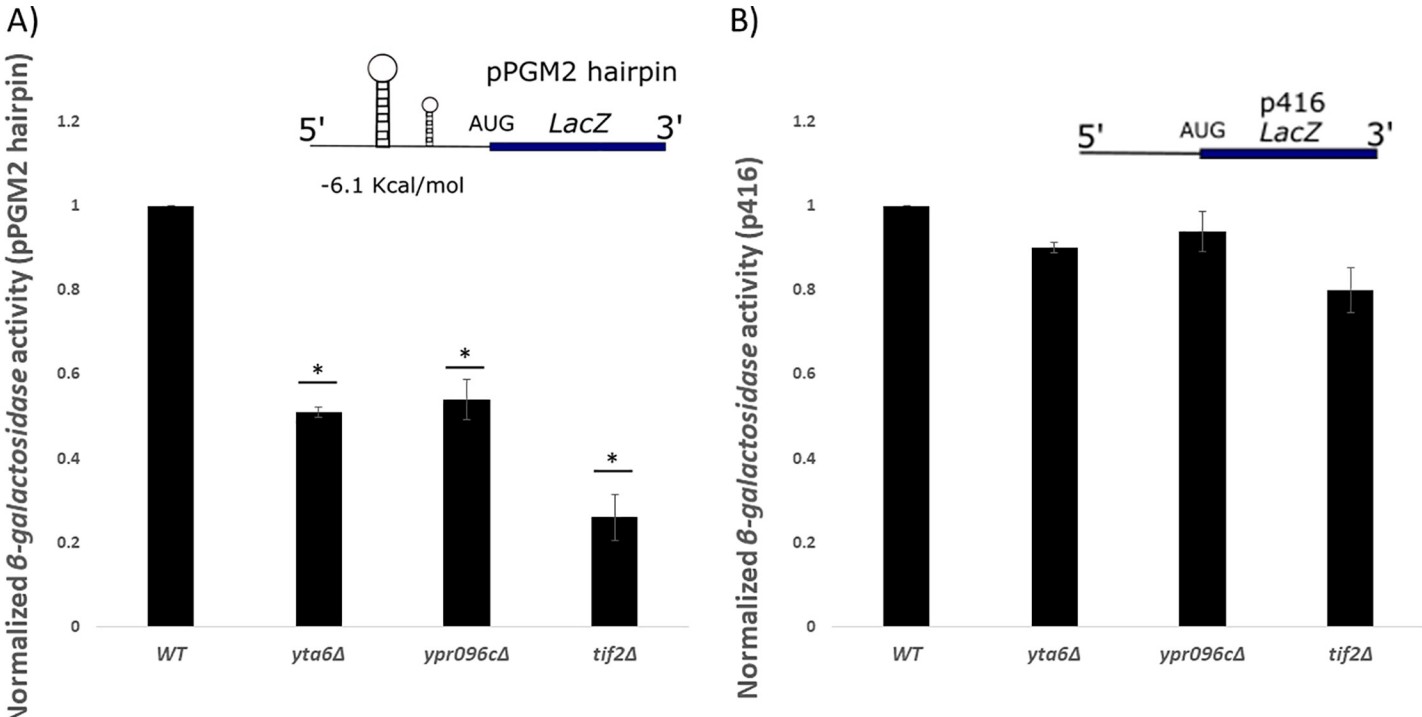

**Fig 4. β-galactosidase expression analysis in different yeast strains.** Activities from *β-galactosidase* mRNAs that carry 5'-UTR of *PGM2* mRNA (pPGM2) (A) upstream of *LacZ* reporter was reduced in *yta6Δ* and *ypr096cΔ* strains; *tif2Δ* was used as a positive control. Strains carrying low complexity regions upstream of *LacZ* reporters p416 (B) did not show as significant reductions in *β-galactosidase* activity. Values are normalized to that for WT which resulted in average *β-galactosidase* values of 38.1U and 407.5U for pPGM2 and p416 constructs, respectively. Each experiment was repeated at least three times (n ≥ 3) and error bars represent standard deviation. * represent statistically significant results (*P*-value ≤ 0.05) compared to the WT. t-test analysis (*P*-value ≤ 0.05) was used to compare differences. The insets represent schematic reporter mRNA structures.

function of genes under a specific condition. The activities of many genes are known to be condition dependent. For example, the expression of many DNA repair genes are regulated in response to DNA damage [42,43]. To this end, we investigated conditional nGIs for *YTA6* and *YPR096C* in the presence of a mild concentration of LiCl (3 mM). Illustrated in Fig 6 *YTA6* formed a total of 14 conditional nGIs. On the basis of their functions and cellular processes in which they participate, these genes can be divided into different categories. Of note, the category of genes involved in protein biosynthesis was the only significantly enriched category (P = 1.6e-4). Within this category, we find 7 genes including, *RPL2B* that encodes large ribosomal subunit protein 2B and *YDR159W* that codes for a protein required for biogenesis of small ribosomal subunit. *YPR096C* formed 13 conditional nGIs, 6 of which belonged to the category of protein biosynthesis (P = 6.6e-4). The genes in this category include *YDL081C* that codes for ribosomal stalk protein P1 alpha and *YER153C* that codes for a mitochondrial translation activator. The conditional nGIs observed here suggest a possible functional association for *YTA6* and *YPR096C* to protein biosynthesis when cells are challenged with LiCl.

Phenotypic Suppression Array (PSA) analysis focuses on another form of GIs, where a specific phenotype associated with a gene deletion mutant is suppressed by the over-expression of the second gene [32,44,45]. This type of GI generally indicates a close functional association where the activity of an over-expressed gene compensates for the absence of the others. To this end, we subjected the gene expression array (described above) to 10 mM of LiCl. In this concentration, a number of strains showed sensitivity. We then attempted to reverse the observed sensitivities by over-expression of either *YTA6* or *YPR096C* in these mutants. Interestingly

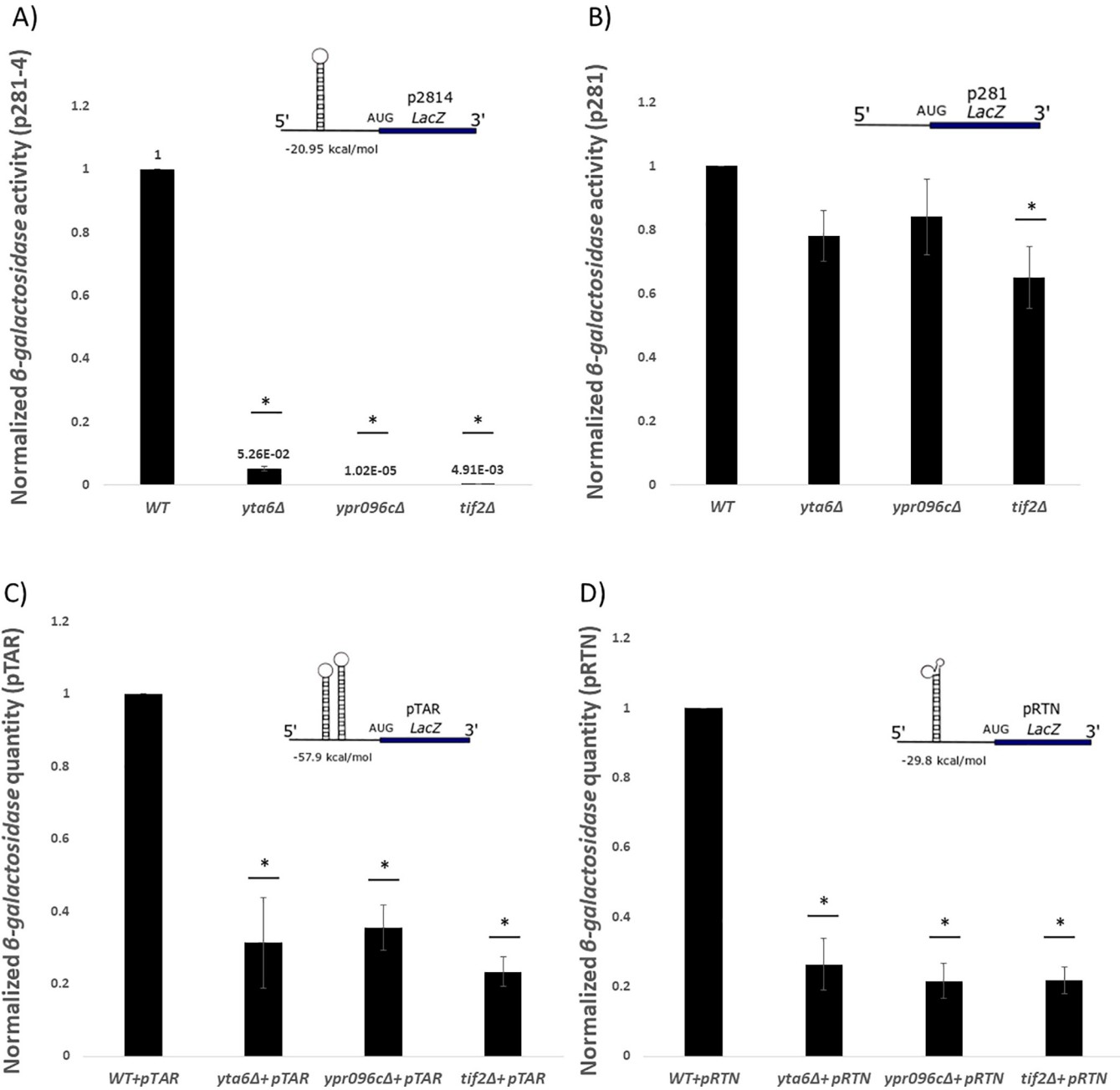

**Fig 5. Normalized *β-galactosidase* activity is lower in *yta6Δ* and *ypr096cΔ* for structured mRNAs.** A strong hairpin structure (p281-4) (A) upstream of a *LacZ* reporter pTAR (C) highly structured 5'-UTR of *HIV1-tar* and pRTN (D) constructs contain the highly structured 5'-UTR of *FOAP-11* genes in front of the *β-galactosidase* reporter mRNA. P281 (B) was served as a control plasmid with no inhibitory structure did not show as significant reductions in *β-galactosidase* activity. Values are normalized to that for WT which resulted in average *β-galactosidase* values of 14.1U and 37.9U for pRTN and p416 constructs, respectively. Each experiment was repeated at least three times (n ≥ 3) and error bars represent standard deviation. * represent statistically significant results (*P*-value ≤ 0.05) compared to the WT. t-test analysis (*P*-value ≤ 0.05) was used to compare differences. The insets represent schematic reporter mRNA structures.

over-expression of either *YTA6* or *YPR096C* compensated for the sensitivity of the same two gene deletions, *bck1Δ* and *eap1Δ*, to LiCl (Fig 7). We confirmed our PSA data using spot test drug sensitivity analysis (Fig 7). We observed that sensitivity of *bck1Δ* and *eap1Δ* to 10 mM LiCl was relieved by introducing *pYTA6* and *pYPR096C* over-expression plasmids into

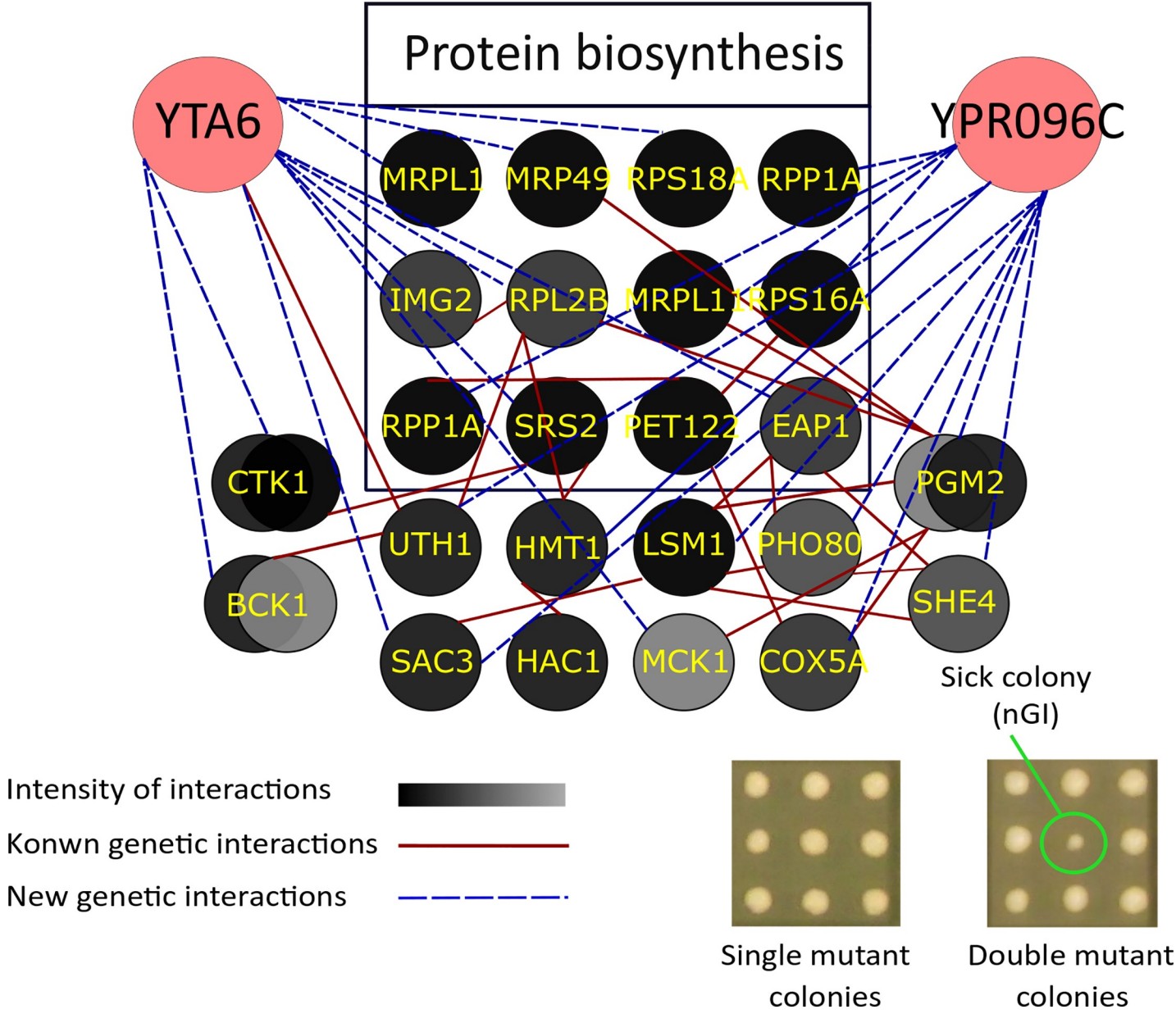

**Fig 6. Conditional nGIs for *YTA6* and *YPR096C* in the presences of 3 mM concentration of LiCl.** Our data shows a cluster of interactors involved in the protein biosynthesis pathway for *YTA6* (P = 1.6e-4) and *YPR096C* (P = 6.6e-4). *CTK1*, *HAC1*, *BCK1*, *MRPL1*, and *PGM2* are mutual hits shared between *YTA6* and *YPR096C*. Circles represent genes, dashed lines represent nGIs identified in this study and solid lines represent previously reported interactions in the literature. The inset represents an example of a typical interaction.

deletion mutant strains (Fig 7). The fact that *YTA6* and *YPR096C* compensated the same two gene deletions, further connects their activities together in the context of LiCl sensitivity. Another possibility is that the over-expression of *YTA6* and *YPR096C* would improve *PGM2* mRNA translation, leading to an increase in *PGM2* activity in the cells that was shown to confer resistance to lithium in galactose medium [11]. According to this hypothesis, if the main cause of toxicity under these conditions is the decrease in *PGM2* activity, the over-expression of *YTA6* and *YPR096C* would be "solving" the original problem and thus making any yeast strain more tolerant to lithium, not only those with a related function in the cell. *Bck1* is

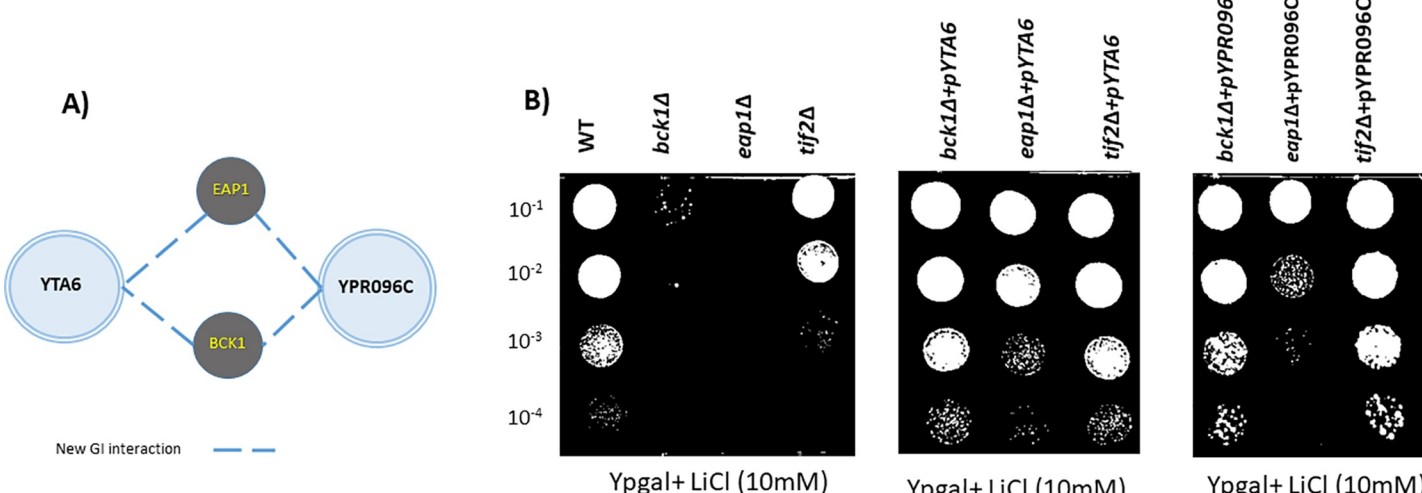

**Fig 7. Over-expression of *YTA6* and *YPR096C* compensate for the sensitivity of *eap1Δ* and *bck1Δ* to 10 mM LiCl.** (A) *BCK1* and *EAP1* are known to be involved in translation initiation via *DHH1* through previously reported genetic. New genetic interactions (PSA-based) identified in this study are shown with dashed lines. (B) Spot test analysis confirms the relief of drug sensitivity to LiCl for *eap1Δ and bck1Δ* by over-expression of *YTA6* and *YPR096C*. Spot test analysis was repeated three times (n = 3) with similar outcomes.

reported to function in cell wall integrity pathway and deadenylation of mRNAs and *Eap1* is an eIF4E-associated protein and accelerates the decapping of mRNAs. They have both been implemented in the regulation of alternative translation initiation via *Dhh1p*, a helicase protein [46–49]. *Dhh1p* is a member of the DEAD-box family of RNA helicases capable of unwinding strong secondary structures. It functions in mRNA decapping and translational repression among other processes [45,50]. A proposed functional association for both *YTA6* and *YPR096C* to the regulation of translation via *Dhh1* merits further investigations.

## Supporting information

**S1 Fig. Drug sensitivity analysis for different yeast strains on YPD media.** No increased LiCl sensitivity was observed for deletion mutant strains for *YTA6* and *YPR096C* in media containing glucose as a carbon source. Spot test analysis was repeated at least three times (n ≥ 3) with similar outcomes.
(TIF)

**S2 Fig. The secondary structure of *PGM2* 5'-UTR.** Unlike most yeast ORFs, the 5' UTR of PGM2 is thought to be structured (38).
(TIF)

**S1 Table. qRT-PCR raw data for different strains with and without LiCl treatment.** Each experiment was repeated at least three times (n ≥ 3).
(DOCX)

**S2 Table. List of mutant strains in gene expression and random arrays.**
(DOCX)

**S3 Table. List of negative genetic interactions (nGIs) for YTA6 and YPR096C (no LiCl in media).**
(DOCX)

**S1 Raw Images.**
(PDF)

## Acknowledgments

This work is dedicated to the memory of our friend and colleague Fareed Arasteh who lost his life in the Tehran Plane Crash, 2020.

## Author Contributions

**Conceptualization:** Mohsen Hooshyar, Ashkan Golshani.

**Data curation:** Maryam Hajikarimlou, Tom Kazmirchuk, Sarah Takallou, Narges Zare, Sasi Kumar Jagadeesan, Nathalie Puchacz, Ashkan Golshani.

**Formal analysis:** Maryam Hajikarimlou, Houman Moteshareie, Mohsen Hooshyar, Sarah Takallou, Bahram Samanfar, Ashkan Golshani.

**Funding acquisition:** Ashkan Golshani.

**Investigation:** Maryam Hajikarimlou, Houman Moteshareie, Tom Kazmirchuk, Myron Smith, Martin Holcik.

**Methodology:** Maryam Hajikarimlou, Katayoun Omidi, Sarah Shaikho.

**Project administration:** Ashkan Golshani.

**Supervision:** Mohan Babu, Myron Smith, Martin Holcik, Bahram Samanfar, Ashkan Golshani.

**Validation:** Maryam Hajikarimlou, Daniel Burnside, Narges Zare, Ashkan Golshani.

**Visualization:** Maryam Hajikarimlou.

**Writing – original draft:** Maryam Hajikarimlou, Ashkan Golshani.

**Writing – review & editing:** Katayoun Omidi, Daniel Burnside, Mohan Babu, Myron Smith, Martin Holcik, Bahram Samanfar, Ashkan Golshani.

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
