## [Decision Letter · Decision Letter 0]

21 Aug 2019

PONE-D-19-21480

Lithium chloride toxicity is connected to regulation of gene expression in yeast

PLOS ONE

Dear Dr. Golshani,

Thank you for submitting your manuscript to PLOS ONE. After careful consideration, we feel that it has merit but does not fully meet PLOS ONE’s publication criteria as it currently stands. Therefore, we invite you to submit a revised version of the manuscript that addresses the points raised during the review process.

As Reviewer 2 points out, LiCl also has an interaction in galactose metabolism, independent of toxicity. The consideration of galactose metabolism intermediates that are likely accumulate in yeast cells creating a large amount of phenotypes may be due to specific galactose-1-phosphate accumulation, already known as part of toxicity of  galactosemia. Since there is a second pathway that has not been considered, discussion of that pathway is required. Add appropriate references.Testing of the alternative pathway per the suggestions of Reviewer 2 is required: 1) control experiments testing lithium toxicity in using media that use other carbon sources (e.g. glucose, glycerol, lactate, etc); 2) test the effect of the gal1 gene deletion on the growth tests of the mutant trains. Test the alternative hypothesis by suppression through deletion of YTA6 and YPR096C.Cell size as an assay of fitness is flawed since it could be due to a higher cell death rate or a slower growth rate. These paramaters need to be determined independently.Present the results of non-treated cells.The PGM2 hairpin hypothesis requires more controls including  testing a) the effect of the hairpin on translational assays and b) the absence of the hairpin on the suppression of the phenotype.The authors need to clarify the potential of non-isogenicity on the nGI results.Statistical analysis requires a description in methods.Figure 6 is difficult to see (see the AE comments below). The major textual problem is the over-comparison with the use of Li in bi-polar disorder, as noted by Reviewer 1.  This may be mentioned only briefly as it is peripheral to the actual studyReviewer 2 has more specialized knowledge essential for the interpretation of the data. Reviewer 1 focused on the presentation.The AE reading of the manuscripts found numerous additional problem. First, Figures 1 and 7 show spot assays that are strips of data. The nature of the splicing is not mentioned in the text and it is unknown whether they are derived from one or more plate. The original data must be presented to the Reviewers for all of the data by the new PLOS One data presentation rules. Second, the other Figures would be assisted by a diagram of the assay shown in that Figure. Third, the statistical tests frequire exact n values and methods in the Figure Legends. Fourth, are the cells that are the origin of the quantitative cell counting independent colonies from independent cells?. If not, this must be repeated with colonies from independent cells that may require a larger sample size.  Fifth, when data are non-statistically significant, they should not be shown in any Figure, including the supplement. Sixth, terms such as gene expression and protein translation are muddled. These need to be fixed. Seventh, the references are incomplete. Please correct carefully. Eighth,  spelling errors detract from the coherent reading of the manuscript.  The authors should have an independent individual unassociated with the paper read the manuscript for spelling and grammar.

We would appreciate receiving your revised manuscript by Oct 05 2019 11:59PM. To enhance the reproducibility of your results, we recommend that if applicable you deposit your laboratory protocols in protocols.io, where a protocol can be assigned its own identifier (DOI) such that it can be cited independently in the future. For instructions see: http://journals.plos.org/plosone/s/submission-guidelines#loc-laboratory-protocols

We look forward to receiving your revised manuscript.

Kind regards,

Arthur J. Lustig, PhD

Academic Editor

PLOS ONE

Journal Requirements:

Reviewers' comments:

Reviewer's Responses to Questions

**Comments to the Author**

1. Is the manuscript technically sound, and do the data support the conclusions?

Reviewer #1: Yes

Reviewer #2: Partly

2. Has the statistical analysis been performed appropriately and rigorously? 

Reviewer #1: Yes

Reviewer #2: No

3. Have the authors made all data underlying the findings in their manuscript fully available?

Reviewer #1: Yes

Reviewer #2: No

4. Is the manuscript presented in an intelligible fashion and written in standard English?

Reviewer #1: Yes

Reviewer #2: Yes

5. Review Comments to the Author

Reviewer #1: This is a concise piece of work on a physiological target function of LiCl toxicity in yeast. The experiments are properly conducted and the results are reasonably interpreted. The presentation is, however, somewhat misleading. More than half (six lines out of ten) of the Abstract describes the link between LiCl and bipolar disorder, although the work has very little to do with the disease. This disproportional presentation is confusing and misleading for general readership. It may be acceptable to mention bipolar disorder in the Introduction, although I recommend to curtail the part from the Abstract.

Typos or errors:

l.82

Insert a space after his5.

l.130, l.166,

Capitalize c of Licl.

l.180-188

This section should be in the Introduction, and therefore should be removed.

l.191

TIF2 should be Italicized.

l.201

Insert a space between 10 and mM.

l.218 and Fig.1 and 7

Ypgal or YPgal (l.107)? Stick to the same nomenclature.

l.356

EAP1, not EAp1.

l.356

The legend mentions dhh1D but the corresponding strain is tif2D in the Fig.7. Which is correct?

Reviewer #2: The present work aims to describe new molecular mechanisms of lithium since it has been used in treatment of bipolar disorder for decades without a complete knowledge of it mechanism of action. Using a yeast genetic approach, the authors identified two genes that when deleted in yeast cells decreases tolerance to lithium treatment in the presence of galactose. Moreover, the deletion of these genes - YTA6 and YPR096C - seems to enhance lithium toxicity by interfering with PGM2 translation, an already known target of lithium chloride. Further, as PGM2 mRNA possesses a highly structured hairpin in the 5’ UTR region, the authors were able to demonstrate that YTA6 and YPR096C participate in the process of translation of some other highly structured mRNAs, suggesting a possible mechanism explaining why PGM2 translation and cell growth are affected by knocking out those genes in the presence of LiCl. As YTA6 and YPR096C are yeast genes with little information so far, the authors focused in genetic interaction screenings to further identify how those deletions could promote emerging double-knockout phenotypes when crossed with deletion mutants of genes involved in gene expression. Using this tool, they could observe enrichment in negative genetic interactions with genes already known to participate in protein biosynthesis. Moreover, by overexpressing YTA6 and YPR096C in two of the negative interaction hits of the previous screening – bck1� and eap1� -, they observed suppression of LiCl toxicity in YPGal medium.

The data presented by the authors contribute to the literature in the context of LiCl toxicity in galactose containing medium, and also propose a function for two yeast genes that, until now, little was known. But, in my point of view, major concerns have to be addressed to fit criteria for publication.

Major concerns:

- The authors argue that effects observed by LiCl treatment regards only lithium toxicity but, in fact, Masuda et al 2001,2008 demonstrated that there is a potent interaction of LiCl treatment and galactose metabolism. By blocking PGM2 activity with lithium, galactose metabolism intermediates accumulate in yeast cells and a large amount of phenotypes are due to specific galactose-1-phosphate accumulation, a molecule already known as part of toxicity of a genetic disease called classic galactosemia. We cannot exclude the hypothesis presented by the authors that lithium is (directly) inducing a protein translation problem, but at this point they cannot exclude that the translation problem is being induced by galactose-1-phosphate (or other intermediary metabolite of the galactose pathway) accumulation neither. No comment about the galactose-1-phosphate toxicity was presented in this manuscript. Because previous work has observed that most phenotypes under these condition can be suppressed by the galactokinase gene GAL1 deletion, authors should at least comment on the hypothesis, but preferably, test it. Suggestions for this test are: 1) control experiments testing lithium toxicity in other media containing other carbon sources (e.g. glucose, glycerol, lactate, etc); 2) test the effect of the gal1 gene deletion on the growth tests of the mutant trains. If the deletion of YTA6 and YPR096C can be completely suppressed by gal1 deletion, it would favor the hypothesis that the toxicity modulated by the presence of these genes is more related to galactose-1-phosphate accumulation than to lithium toxicity directly.

- In order to claim that the effect of the deletions of YTA6 and YPR096C on lithium/galactose toxicity is really due to the impact on PGM2 translation, I would suggest authors to: 1) test the actual PGM2 mRNA hairpin on the translational assays used in this work; 2) test the expression of a PGM2 gene allele without the hairpin – I would expect that this allele would suppress the effect of the deletions.

- The quality of the presented figures is bad, some are almost impossible to visualize (especially figure 6), please increase quality for publication.

- Usually, yeast spot growth assays are presented as one photograph of each plate containing all the strains that are to be compared. It makes the comparison of relative growth rate more straight-forward and convincing.

- The nGI screening present a bias as authors crossed the mutants yta6� and ypr0963c� to a called “gene expression library”, thus enhancing the chances of enrichment in protein biosynthesis interactions. So, I am not sure whether the enrichment of the class observed is a good indicator in this case. Also, since the screening was performed with a subset of the entire library, authors should list all the mutants included in the screening.

- Some of the arguments used during the manuscript are missing references, and some are wrongly interpreted from the literature. For example, in the lines 35/36 authors argue that LiCl reduces PGM2 expression but Masuda et al 2001 shows that lithium treatment increases the mRNA levels of PGM2. In lines 255-257 the group argue that deletion of TIF2 reduces PGM2 expression in response to LiCl accordingly to Montero-lomeli et al., 2002 but this data does not exist in this publication.

- Please recheck the references section of your work. Many references are lacking informations like journal name and/or DOI. In line 173 there is a reference missing from the list (Memarian et al., 2007). In line 191 the reference Bro et al., 2003 is actually Montero-Lomeli et al., 2003.

- No statistical analysis details are presented in the work. Although differences in data are clear, please indicate in the method section the type of test used in Figure 4 and perform statistical analysis for the rest of the data.

Minor concerns:

- It would be interesting if authors could present the result of non-treated cells on figure 3b to observe any effect of the deletions on the basal expression of PGM2 gene.

- Because of the way the quantitative growth experiment was performed, it is impossible to discern whether the effect of the deletion of YTA6 and YPR096C genes is increasing lethality (cytotoxicity) or decreasing growth rates (cytostatic effect). It would be interesting to discern between these effects performing some viability assays since the figure 2 result shows that overexpression of genes lead to a colony number higher than the WT strain, suggesting a decreased cytotoxicity.

- The term gene expression is usually used to address gene transcription, not so much for effect in translation. Although I do not consider the actual (in this manuscript) use of the term wrong, I suggest being more precise in describing the phenotypes observed, even in the title, and better establish that the hypothesis is that these genes (and lithium?) affect the process of translation.

- For better understanding, improve the description of the method for colony counting in the methods section.

- The housekeeping gene used to normalize qRT-PCR is always an issue, and I this case PGK1 was used. Is this a good housekeeping gene to this context? Have you tried others such as ACT1 or TAF10?

- Add references to the arguments in lines 41-42, 42-43, 265 (for TIF2 as control),348-349.

- Period between lines 47-52 is difficult to understand.

- Misspelling in lines 126 (Reference of krogan et al 2003); 227 (MasudA et al 2001); 245 – legend- PGM2 protein and mRNA content analysis

- Cite Tong et al., 2001 also in the methods section

6. PLOS authors have the option to publish the peer review history of their article (what does this mean?). If published, this will include your full peer review and any attached files.

Reviewer #1: No

Reviewer #2: Yes: Claudio Akio Masuda

---

## [Author Response · Author response to Decision Letter 0]

5 Feb 2020

Responses to Editor and Reviewers comments

We would like to start by thanking Drs. AJ Lusting, CA Masuda and unanimous reviewer #1 for their invaluable time and comments to improve the quality of this manuscript. 

Editor’s comments:

• As Reviewer 2 points out, LiCl also has an interaction in galactose metabolism, independent of toxicity. The consideration of galactose metabolism intermediates that are likely accumulate in yeast cells creating a large amount of phenotypes may be due to specific galactose-1-phosphate accumulation, already known as part of toxicity of galactosemia. Since there is a second pathway that has not been considered, discussion of that pathway is required. Add appropriate references.

Our response: The text is now modified accordingly. 

• Testing of the alternative pathway per the suggestions of Reviewer 2 is required: 1) control experiments testing lithium toxicity in using media that use other carbon sources (e.g. glucose, glycerol, lactate, etc); 2) test the effect of the gal1 gene deletion on the growth tests of the mutant trains. Test the alternative hypothesis by suppression through deletion of YTA6 and YPR096C.

Our response: Both suggested experiments are now performed and the results are reported. Please refer to our answer to reviewer #2 major comment #1.

• Cell size as an assay of fitness is flawed since it could be due to a higher cell death rate or a slower growth rate. These paramaters need to be determined independently.

Our response: We agree with this comment. Although cell death and growth rate can both represent sensitivity, they do not signify the same parameter. In the current study, general cell sensitivity serves as a starting point for follow-up analyses. We feel the outcome of this experiment will not affect the conclusions of the current study.

• Present the results of non-treated cells.

Our response: The text is now modified accordingly (Fig 3). 

• The PGM2 hairpin hypothesis requires more controls including testing a) the effect of the hairpin on translational assays and b) the absence of the hairpin on the suppression of the phenotype.

Our response: Suggested experiments are now performed. We now have a new b-gal construct with and without PGM2 hairpin. The results are shown in modified Figure 4 panels A and B.

• The authors need to clarify the potential of non-isogenicity on the nGI results.

Our response: We agree that such screens have inherent bias. Please refer to our answer to the same point raised by Reviewer #2, under major concerns, point 5. We have used a random plate to estimate the overall rate of GIs for a target gene in our hands. In this case we can drive more meaningful P-values. For clarity, the description of the control plate is modified. 

• Statistical analysis requires a description in methods.

Our response: The text is now modified accordingly.

• Figure 6 is difficult to see (see the AE comments below). 

Our response: The resolution is now improved.

• The major textual problem is the over-comparison with the use of Li in bi-polar disorder, as noted by Reviewer 1. This may be mentioned only briefly as it is peripheral to the actual study

Our response: The text is now modified accordingly.

• Reviewer 2 has more specialized knowledge essential for the interpretation of the data. Reviewer 1 focused on the presentation.

The AE reading of the manuscripts found numerous additional problem. First, Figures 1 and 7 show spot assays that are strips of data. The nature of the splicing is not mentioned in the text and it is unknown whether they are derived from one or more plate. The original data must be presented to the Reviewers for all of the data by the new PLOS One data presentation rules. Second, the other Figures would be assisted by a diagram of the assay shown in that Figure. Third, the statistical tests frequire exact n values and methods in the Figure Legends. Fourth, are the cells that are the origin of the quantitative cell counting independent colonies from independent cells?. If not, this must be repeated with colonies from independent cells that may require a larger sample size. Fifth, when data are non-statistically significant, they should not be shown in any Figure, including the supplement. Sixth, terms such as gene expression and protein translation are muddled. These need to be fixed. Seventh, the references are incomplete. Please correct carefully. Eighth, spelling errors detract from the coherent reading of the manuscript. The authors should have an independent individual unassociated with the paper read the manuscript for spelling and grammar.

Our response: The text is now modified accordingly. We have kept some of the non-statistically significant data, only when a comparison is needed. For example, growth under different conditions. 

Reviewers’ Comments to the Author

Reviewer #1: This is a concise piece of work on a physiological target function of LiCl toxicity in yeast. The experiments are properly conducted and the results are reasonably interpreted. The presentation is, however, somewhat misleading. More than half (six lines out of ten) of the Abstract describes the link between LiCl and bipolar disorder, although the work has very little to do with the disease. This disproportional presentation is confusing and misleading for general readership. It may be acceptable to mention bipolar disorder in the Introduction, although I recommend to curtail the part from the Abstract.

Our response: The abstract and the text is modified accordingly. 

Typos or errors:

l.82

Insert a space after his5.

l.130, l.166,

Capitalize c of Licl.

l.180-188

This section should be in the Introduction, and therefore should be removed.

l.191

TIF2 should be Italicized.

l.201

Insert a space between 10 and mM.

l.218 and Fig.1 and 7

Ypgal or YPgal (l.107)? Stick to the same nomenclature.

l.356

EAP1, not EAp1.

l.356

The legend mentions dhh1D but the corresponding strain is tif2D in the Fig.7. Which is correct?

Our response: The text is modified accordingly.

Reviewer #2: 

Major concerns:

1- The authors argue that effects observed by LiCl treatment regards only lithium toxicity but, in fact, Masuda et al 2001,2008 demonstrated that there is a potent interaction of LiCl treatment and galactose metabolism. By blocking PGM2 activity with lithium, galactose metabolism intermediates accumulate in yeast cells and a large amount of phenotypes are due to specific galactose-1-phosphate accumulation, a molecule already known as part of toxicity of a genetic disease called classic galactosemia. We cannot exclude the hypothesis presented by the authors that lithium is (directly) inducing a protein translation problem, but at this point they cannot exclude that the translation problem is being induced by galactose-1-phosphate (or other intermediary metabolite of the galactose pathway) accumulation neither. No comment about the galactose-1-phosphate toxicity was presented in this manuscript. Because previous work has observed that most phenotypes under these condition can be suppressed by the galactokinase gene GAL1 deletion, authors should at least comment on the hypothesis, but preferably, test it. Suggestions for this test are: 1) control experiments testing lithium toxicity in other media containing other carbon sources (e.g. glucose, glycerol, lactate, etc); 2) test the effect of the gal1 gene deletion on the growth tests of the mutant trains. If the deletion of YTA6 and YPR096C can be completely suppressed by gal1 deletion, it would favor the hypothesis that the toxicity modulated by the presence of these genes is more related to galactose-1-phosphate accumulation than to lithium toxicity directly.

Our response: We completely agree with this comment. It is a big oversight from our end. We have now performed both suggested experiments. The results are shown in Figure 1D and Figure 4 A and B. In brief, we believe that the influence of LiCl on translation in through previously reported galactose metabolism. The text is modified accordingly. Considering this comment and the new data, the title of the manuscript is now modified. 

2- In order to claim that the effect of the deletions of YTA6 and YPR096C on lithium/galactose toxicity is really due to the impact on PGM2 translation, I would suggest authors to: 1) test the actual PGM2 mRNA hairpin on the translational assays used in this work; 2) test the expression of a PGM2 gene allele without the hairpin – I would expect that this allele would suppress the effect of the deletions.

Our response: We now have a new b-gal reporter construct with and without PGM2 hairpin. The results are shown in new Figure 4A and B. Reduction of b-gal is linked to the presence of the PGM2 hairpin on the mRNA.

3- The quality of the presented figures is bad, some are almost impossible to visualize (especially figure 6), please increase quality for publication.

Our response: The manuscript is modified accordingly.

4- Usually, yeast spot growth assays are presented as one photograph of each plate containing all the strains that are to be compared. It makes the comparison of relative growth rate more straight-forward and convincing.

Our response: The new Figure 1 is modified accordingly.

5- The nGI screening present a bias as authors crossed the mutants yta6� and ypr0963c� to a called “gene expression library”, thus enhancing the chances of enrichment in protein biosynthesis interactions. So, I am not sure whether the enrichment of the class observed is a good indicator in this case. Also, since the screening was performed with a subset of the entire library, authors should list all the mutants included in the screening.

Our response: We agree with this comment. Although this type of enriched screening and its modified versions have been commonly used by us (e.g. Alamgir et al BMC Genomics, 2008; Samanfar et al Mol Biosyst, 2013; Gagarinova et al Cell Rep, 2016; etc) and others (e.g. Laribee et al 2007, PNAS; Collins et al Nature, 2007; Roguev et al Science, 2008; Zheng et al Mol Syst Biol, 2010; etc) they all have certain inherent bias associated with them. In our case we always use a random set of mutants (384 strains) as a control plate to estimate the overall rate of GIs for a target gene in our hands. In this case we can drive more meaningful P-values. For clarity, the description of the control plate is modified. 

6- Some of the arguments used during the manuscript are missing references, and some are wrongly interpreted from the literature. For example, in the lines 35/36 authors argue that LiCl reduces PGM2 expression but Masuda et al 2001 shows that lithium treatment increases the mRNA levels of PGM2. In lines 255-257 the group argue that deletion of TIF2 reduces PGM2 expression in response to LiCl accordingly to Montero-lomeli et al., 2002 but this data does not exist in this publication.

Our response: The manuscript is modified accordingly.

7- Please recheck the references section of your work. Many references are lacking informations like journal name and/or DOI. In line 173 there is a reference missing from the list (Memarian et al., 2007). In line 191 the reference Bro et al., 2003 is actually Montero-Lomeli et al., 2003.

Our response: The manuscript is modified accordingly.

8- No statistical analysis details are presented in the work. Although differences in data are clear, please indicate in the method section the type of test used in Figure 4 and perform statistical analysis for the rest of the data.

Our response: The manuscript is modified accordingly.

Minor concerns:

- It would be interesting if authors could present the result of non-treated cells on figure 3b to observe any effect of the deletions on the basal expression of PGM2 gene.

Our response: The suggested experiment is performed and now included in Figure 3. 

- Because of the way the quantitative growth experiment was performed, it is impossible to discern whether the effect of the deletion of YTA6 and YPR096C genes is increasing lethality (cytotoxicity) or decreasing growth rates (cytostatic effect). It would be interesting to discern between these effects performing some viability assays since the figure 2 result shows that overexpression of genes lead to a colony number higher than the WT strain, suggesting a decreased cytotoxicity.

Our response: We agree with this comment. In the current study, general cell sensitivity serves as a starting point for follow-up analyses. We feel the outcome of this experiment will not affect the conclusions of the current study, but plan to include it in our future work. Also we discovered an issue with normalization of our overexpression genes in Figure 2; this is now corrected. 

- The term gene expression is usually used to address gene transcription, not so much for effect in translation. Although I do not consider the actual (in this manuscript) use of the term wrong, I suggest being more precise in describing the phenotypes observed, even in the title, and better establish that the hypothesis is that these genes (and lithium?) affect the process of translation.

Our response: The manuscript is modified accordingly. The title is also modified.

- For better understanding, improve the description of the method for colony counting in the methods section.

Our response: The manuscript is modified accordingly.

- The housekeeping gene used to normalize qRT-PCR is always an issue, and I this case PGK1 was used. Is this a good housekeeping gene to this context? Have you tried others such as ACT1 or TAF10?

Our response: This is an interesting point. We have not used additional housekeeping genes but would be interesting to examine a second gene in future.

- Add references to the arguments in lines 41-42, 42-43, 265 (for TIF2 as control),348-349.

Our response: The manuscript is modified accordingly.

- Period between lines 47-52 is difficult to understand.

Our response: The manuscript is modified accordingly.

- Misspelling in lines 126 (Reference of krogan et al 2003); 227 (MasudA et al 2001); 245 – legend- PGM2 protein and mRNA content analysis

Our response: The manuscript is modified accordingly.

- Cite Tong et al., 2001 also in the methods section

Our response: The manuscript is modified accordingly.

---

## [Decision Letter · Decision Letter 1]

3 Mar 2020

PONE-D-19-21480R1

Sensitivity of yeast to lithium chloride connects the activity of YTA6 and YPR096C to translation of structured mRNAs

PLOS ONE

Dear Dr. Golshani,

Thank you for submitting your manuscript to PLOS ONE. After careful consideration, we feel that it has merit but does not fully meet PLOS ONE’s publication criteria as it currently stands. Therefore, we invite you to submit a revised version of the manuscript that addresses the points raised during the review process.

The AE agrees with the comments of Reviewers 1 and 2. In particular, Reviewer 2 provides a number of appropriate issues of confusion that results from both the manner of presentation and some bona fide confusion regarding the pathway involvement.The AE finds that the manuscript needs to be written in a more accessible fashion and with greater clarity.  These include methodological, logical and data presentation issues.   In particular, in order of priority,The manuscript has a large number of grammatical mistakes;. in particular, the lack of use of articles ( e.g, "the"). This problem is present throughout the paper and compromises the ease of reading the manuscript.  This is necessary to meet the standards for publication in PLOS. Please have an individual familiar with publication standards review your grammar.The methodology lacks sufficient clarity for reproduction. This includes an insufficient description of selection schemes, strain development, and design rationale.Specific errors include the definition of NAT (it is not nourseothricin but, rather nourseothricin N-acetyl transferase).How was the concentration of LiCl determined? What were the effects in different concentration of LiCl> and a clear relationship between context for which methods are involved in which experiments. Is the statistical basis of the qPCR the MIQE convention? If not, please redo these evaluations according the this convention.Define SDL.Statistical tests should include the sets of values that are being compared (for example in Western blot analysis). Also place n value and statistical test in Figure Legends.Please define the unconventional use of "fitness". Are you not referring to colony size?Please describe the tests for the functionality of fusion proteins used in this study. Also clarify which experiments used and did not use the fusion protein.The introduction of high copy number genes that complement nulls is not a test of complementation of the gene. Rather, re-integration of a single copy gene is needed to prove that the correct gene has been isolated in plasmids. Much of the data is presented graphically in Figures 2,3,4,5. An example of each dataset is required in these Figures. Similarly, a subset of the data interactions summarized in Figure 6 that have been uncovered in this paper should be presented. In addition, in a Supplementary Data section, by PLOS One regulations, all of the raw data must be presented so that the data is can be directly assessed. In the spot assays in Figure 1, are all the plates derived from the same microarray plate?The logic behind the interpretation of mutations in the presence of gal1 null alleles must be explained more clearly as noted by Reviewer 2.The argument for hairpin structure involvement is quite good. Why was the relationship to the growth sensitivity data not tested by conducting hairpin structure experiments in gal null alleles in a subset of experiments?The isolation of interacting genes in some ribosomal protein genes and regulators of translation implicates  the translation process. However,  conclusions the basis of these studies should be in line with the strength of the data.Eliminate the bipolar discussion in the Results.  It is not relevant to this manuscript.There are no conflicts between the Reviewers. Reviewer 1 is primarily concerned with presentation issues. Reviewer 2 is involved with scientific issues and mis-statements and the AE is examining the manuscript as an entirety.  All of the issues must be addressed.

We would appreciate receiving your revised manuscript by Apr 17 2020 11:59PM. To enhance the reproducibility of your results, we recommend that if applicable you deposit your laboratory protocols in protocols.io, where a protocol can be assigned its own identifier (DOI) such that it can be cited independently in the future. For instructions see: http://journals.plos.org/plosone/s/submission-guidelines#loc-laboratory-protocols

We look forward to receiving your revised manuscript.

Kind regards,

Arthur J. Lustig, PhD

Academic Editor

PLOS ONE

Additional Editor Comments (if provided):

Further work is needed in presentation, sufficiency of data, control experiments, quality of methodology, and in experimental interpretation.

Reviewers' comments:

Reviewer's Responses to Questions

**Comments to the Author**

1. If the authors have adequately addressed your comments raised in a previous round of review and you feel that this manuscript is now acceptable for publication, you may indicate that here to bypass the “Comments to the Author” section, enter your conflict of interest statement in the “Confidential to Editor” section, and submit your "Accept" recommendation.

Reviewer #1: (No Response)

Reviewer #2: (No Response)

2. Is the manuscript technically sound, and do the data support the conclusions?

Reviewer #1: Yes

Reviewer #2: Yes

3. Has the statistical analysis been performed appropriately and rigorously? 

Reviewer #1: I Don't Know

Reviewer #2: Yes

4. Have the authors made all data underlying the findings in their manuscript fully available?

Reviewer #1: Yes

Reviewer #2: No

5. Is the manuscript presented in an intelligible fashion and written in standard English?

Reviewer #1: Yes

Reviewer #2: Yes

6. Review Comments to the Author

Reviewer #1: As I pointed out in the previous review, the work has very little to do with BD. Furthermore, the results and discussion section should not contain lengthy explanation of the background information. From these reasons, I recommend to eliminate the section from l.174 to l.185.

Reviewer #2: The authors addressed most of the concerns raised in the first round of revision and the manuscript improved significantly. However, I still think there are a few point that should be corrected in order to make the manuscript ready for publication.

1) Abstract: “Reduced activity of phosphoglucomutase in the presence of galactose causes an accumulation of glucose-1-p leading to a number of phenotypes including growth defect.”

Glucose-1-phosphate is not the only metabolite accumulated under this condition. So, it is more accurate to say that “intermediate metabolites of galactose metabolism” accumulates.

And in work of other groups and ours, many of the phenotypes induced by PGM inhibition is mimicked by the deletion of the GAL7 gene. Because of that, we suggest that galactose-1-phosphate (not glucose-1-phosphate) may be the real culprit of these effects. (Reasoning: Galactose-1-phosphate is the only common metabolite downstream of galactokinase (GAL1) that accumulate in both PGM inhibition and GAL7 inhibition conditions)

2) Abstract: “In the current study we identify two understudied genes, YTA6 and YPR096C that when deleted increase cell sensitivity to LiCl treatment in yeast.”

Specify that the increased sensitivity to lithium is observed only in the presence of galactose.

3) Abstract: “…YTA6 and YPR096C exert their activities by influencing PGM2 at the level of translation.”

Suggest to highlight that the mRNA of PGM2 has a structured 5’ UTR to match the title.

4) “Lithium chloride (LiCl) has remained an important treatment option for BD for more than ten decades (2,3).”

Correct “ten decades”. The lithium use to treat neurological disorders started with the work of Dr. John Cade in 1949 and only “officially” accepted later on.

5) “Phosphoglucomutase is responsible for converting glucose-1-phosphate to glucose-6-phosphate.”

Suggestion: add here the information that lithium is an inhibitor of the enzymatic activity of PGM.

6) “When galactose is used as the carbon source, inhibition of phosphoglucomutase by LiCl results in accumulation of glucose-1-phosphate that in turn causes growth defects (11,12).”

Change glucose-1-phosphate to “galactose metabolites intermediates”.

7) “…be a rapid loss of ribosomal protein gene (RBG) pre-mRNAs”

RBG is not used afterwards.

8) “…LiCl reduces the activity of phosphoglucomutase enzyme leading to the accumulation of galactose-1-phosphate, a toxic intermediate in galactose metabolism.”

Suggestion to change for: … accumulation of intermediate metabolites from the galactose metabolism including galactose-1-phosphate, a toxic intermediate.

9) “we generated double gene deletions for YTA6 or YPR096C with GAL1 gene. Deletion of GAL1 gene relieved the sensitivity of gene deletion mutants for YTA6 or YPR096C to LiCl (Fig 1).”

There is no previous reference to the GAL1 gene, what enzyme it encodes and the reasoning behind the GAL1 deletion effect. This experiment needs more contextualization for a non-expert reader to understand these results.

10) “sensitivity of deletions strains for YTA6 or YPR096C to LiCl diminished when glucose was used as a carbon source further connecting the observed LiCl sensitivity for YTA6 and YPR096C deletion strains to galactose metabolism.”

According to the results presented (YPD + 10 mM LiCl), deletions of YTA6 and YPR096C does not cause ANY sensitivity to LiCl in the absence of galactose, not a DIMINISHED sensitivity to galactose.

That said, it would be interesting to check the effect of these deletions to toxic concentrations of LiCl in YPD (100 - 300 mM range). This experiment could either unmask a possible direct effect of lithium independent of galactose metabolism, or further emphasize that the increased sensitivity of these strains to lithium is dependent on its effect on galactose metabolism.

11) Figures 4 and 5 – Because both figures address the same general question (effect of YTA6 and YPR096C deletion on translation efficiency of mRNAs containing structures), I suggest to either join figures 4 and 5, or transfer the figures 4C and D to figure 5 to isolate on Figure 4 the discussion about PGM2 mRNA.

12) Since the nGI and PSA screenings were performed with a subset of the entire Yeast KO library, authors should list all the mutants included in the screenings and identify the groups (gene expression / random groups).

7. PLOS authors have the option to publish the peer review history of their article (what does this mean?). If published, this will include your full peer review and any attached files.

Reviewer #1: No

Reviewer #2: Yes: Claudio A Masuda

---

## [Author Response · Author response to Decision Letter 1]

4 Apr 2020

Responses to Editor and Reviewers comments

We would like to start by thanking Drs. AJ Lusting, CA Masuda and the unanimous reviewer #1 for their invaluable time and comments to improve the quality of this manuscript. 

Editor’s comments:

• The manuscript has a large number of grammatical mistakes; in particular, the lack of use of articles (e.g, "the"). This problem is present throughout the paper and compromises the ease of reading the manuscript. This is necessary to meet the standards for publication in PLOS. Please have an individual familiar with publication standards review your grammar.

Our response: The text is now modified accordingly. 

• Specific errors include the definition of NAT (it is not nourseothricin but, rather nourseothricin N-acetyl transferase).

Our response: NAT is now replaced by clonNAT (nourseothricin sulfate). 

• How was the concentration of LiCl determined? What were the effects in different concentration of LiCl> and a clear relationship between context for which methods are involved in which experiments.

 Our response: Proper references are now included for the starting concentrations of LiCl. Additional explanation is also provided. Description is added for each method connecting it to the corresponding experiment.

• Is the statistical basis of the qPCR the MIQE convention? If not, please redo these evaluations according the this convention.

Our response: Yes, it is. The text is now modified to reflect this.

• Define SDL.

Our response: “SDL” is now removed from the text. 

• Statistical tests should include the sets of values that are being compared (for example in Western blot analysis). Also place n value and statistical test in Figure Legends.

Our response: The text is now modified accordingly.

• Please define the unconventional use of "fitness". Are you not referring to colony size?

Our response: A description is now added in the Materials and methods section. In Genetic Interaction analysis, colony size is commonly used as a measure of fitness. For example: Tong et al 2001 Science; Toufighi et al 2011 Nature Methods; Roguev et al 2018 Cold Spring Harb Protoc; etc. 

• Please describe the tests for the functionality of fusion proteins used in this study. Also clarify which experiments used and did not use the fusion protein.

Our response: PCR analysis and LiCl sensitivity was used to confirm the integrity of Pgm2p-GFP strain. The Material and methods section is modified accordingly. 

• The introduction of high copy number genes that complement nulls is not a test of complementation of the gene. Rather, re-integration of a single copy gene is needed to prove that the correct gene has been isolated in plasmids.

Our response: The text is now modified accordingly.

• Much of the data is presented graphically in Figures 2,3,4,5. An example of each dataset is required in these Figures. Similarly, a subset of the data interactions summarized in Figure 6 that have been uncovered in this paper should be presented. In addition, in a Supplementary Data section, by PLOS One regulations, all of the raw data must be presented so that the data is can be directly assessed.

Our response: An example for each data set is now added. Negative genetic interaction data is now represented in Table S3. An inset for Figure 4A is now included. An inset for Figure 6 is now included. 

• In the spot assays in Figure 1, are all the plates derived from the same microarray plate?

Our response: Yes, each set is spotted on the same plate and grown under the same conditions. 

• The logic behind the interpretation of mutations in the presence of gal1 null alleles must be explained more clearly as noted by Reviewer 2.

Our response: The text is now modified accordingly.

• The argument for hairpin structure involvement is quite good. Why was the relationship to the growth sensitivity data not tested by conducting hairpin structure experiments in gal null alleles in a subset of experiments?

Our response: Gal1 deletion analysis was to study the function of YTA6 and YPR096C in the context of LiCl mode of toxicity. It was performed to functionally link sensitivity of YTA6 and YPR096C to the accumulation of toxic intermediate when galactose was used as a carbon source. This is in contrast to other experiments that were mainly concerned with the expression and translatability of mRNAs. For example, β-galactosidase activity analyses derived from the constructs that carry different hairpin structures on their 5’-UTRs, including that for PGM2, are independent of GAL1 function. They focus on the ability of mutant strains to translate β-galactosidase mRNAs that contain hairpin structures. Similarly, the RNA (qRT-PCR) and protein (western) content analyses were focused on the expression of PGM2 and not its function. Deletion of GAL1 is expected to have no consequence in the outcome of these experiments and would not change the conclusion of the study about translation of structured mRNAs. 

• The isolation of interacting genes in some ribosomal protein genes and regulators of translation implicates the translation process. However, conclusions the basis of these studies should be in line with the strength of the data.

Our response: The text is now modified to reduce the strength of the concluding sentence regarding GI data.

• Eliminate the bipolar discussion in the Results. It is not relevant to this manuscript.

Our response: The text is now modified accordingly.

Reviewers’ Comments to the Author

Reviewer #1: 

1- As I pointed out in the previous review, the work has very little to do with BD. Furthermore, the results and discussion section should not contain lengthy explanation of the background information. From these reasons, I recommend to eliminate the section from l.174 to l.185.

Our response: The text is now modified accordingly. 

Reviewer #2:

1- Abstract: “Reduced activity of phosphoglucomutase in the presence of galactose causes an accumulation of glucose-1-p leading to a number of phenotypes including growth defect.”

Glucose-1-phosphate is not the only metabolite accumulated under this condition. So, it is more accurate to say that “intermediate metabolites of galactose metabolism” accumulates.

And in work of other groups and ours, many of the phenotypes induced by PGM inhibition is mimicked by the deletion of the GAL7 gene. Because of that, we suggest that galactose-1-phosphate (not glucose-1-phosphate) may be the real culprit of these effects. (Reasoning: Galactose-1-phosphate is the only common metabolite downstream of galactokinase (GAL1) that accumulate in both PGM inhibition and GAL7 inhibition conditions) 

Our response: The text is now modified accordingly. 

2- Abstract: “In the current study we identify two understudied genes, YTA6 and YPR096C that when deleted increase cell sensitivity to LiCl treatment in yeast.”

Specify that the increased sensitivity to lithium is observed only in the presence of galactose.

Our response: The text is now modified accordingly. 

3- Abstract: “…YTA6 and YPR096C exert their activities by influencing PGM2 at the level of translation.”

Suggest to highlight that the mRNA of PGM2 has a structured 5’ UTR to match the title.

Our response: The text is now modified accordingly. 

4- “Lithium chloride (LiCl) has remained an important treatment option for BD for more than ten decades (2,3).”

Correct “ten decades”. The lithium use to treat neurological disorders started with the work of Dr. John Cade in 1949 and only “officially” accepted later on.

Our response: The text is now modified accordingly. 

5- “Phosphoglucomutase is responsible for converting glucose-1-phosphate to glucose-6-phosphate.”

Suggestion: add here the information that lithium is an inhibitor of the enzymatic activity of PGM.

Our response: The text is now modified accordingly. 

6- “When galactose is used as the carbon source, inhibition of phosphoglucomutase by LiCl results in accumulation of glucose-1-phosphate that in turn causes growth defects (11,12).”

Change glucose-1-phosphate to “galactose metabolites intermediates”.

Our response: The text is now modified accordingly. 

7- “…be a rapid loss of ribosomal protein gene (RBG) pre-mRNAs”

RBG is not used afterwards.

Our response: The text is now modified accordingly. 

8- “…LiCl reduces the activity of phosphoglucomutase enzyme leading to the accumulation of galactose-1-phosphate, a toxic intermediate in galactose metabolism.”

Suggestion to change for: … accumulation of intermediate metabolites from the galactose metabolism including galactose-1-phosphate, a toxic intermediate.

Our response: The text is now modified accordingly. 

9- “we generated double gene deletions for YTA6 or YPR096C with GAL1 gene. Deletion of GAL1 gene relieved the sensitivity of gene deletion mutants for YTA6 or YPR096C to LiCl (Fig 1).”

There is no previous reference to the GAL1 gene, what enzyme it encodes and the reasoning behind the GAL1 deletion effect. This experiment needs more contextualization for a non-expert reader to understand these results.

Our response: The text is now modified accordingly. Relative explanation has been added.

10- “sensitivity of deletions strains for YTA6 or YPR096C to LiCl diminished when glucose was used as a carbon source further connecting the observed LiCl sensitivity for YTA6 and YPR096C deletion strains to galactose metabolism.”

According to the results presented (YPD + 10 mM LiCl), deletions of YTA6 and YPR096C does not cause ANY sensitivity to LiCl in the absence of galactose, not a DIMINISHED sensitivity to galactose.

That said, it would be interesting to check the effect of these deletions to toxic concentrations of LiCl in YPD (100 - 300 mM range). This experiment could either unmask a possible direct effect of lithium independent of galactose metabolism, or further emphasize that the increased sensitivity of these strains to lithium is dependent on its effect on galactose metabolism.

Our response: The text is now modified accordingly. Also, we subjected the yeast gene deletion strains for YTA6 or YPR096C to 100 mM of LiCl in YPD and observed no increased sensitivity for these mutants in comparison to a control strain. This observation is now reported in the text and the results are shown in Fig S1. 

11- Figures 4 and 5 – Because both figures address the same general question (effect of YTA6 and YPR096C deletion on translation efficiency of mRNAs containing structures), I suggest to either join figures 4 and 5, or transfer the figures 4C and D to figure 5 to isolate on Figure 4 the discussion about PGM2 mRNA.

Our response: New figures are modified accordingly. The new figure 5 now includes the old figure 4C and 4D panels. 

12- Since the nGI and PSA screenings were performed with a subset of the entire Yeast KO library, authors should list all the mutants included in the screenings and identify the groups (gene expression / random groups).

Our response: The list has been added to the supporting information.

---

## [Decision Letter · Decision Letter 2]

23 Apr 2020

PONE-D-19-21480R2

Sensitivity of yeast to lithium chloride connects the activity of YTA6 and YPR096C to translation of structured mRNAs

PLOS ONE

Dear Dr. Golshani,

Thank you for submitting your manuscript to PLOS ONE. After careful consideration, we feel that it has merit but does not fully meet PLOS ONE’s publication criteria as it currently stands. Therefore, we invite you to submit a revised version of the manuscript that addresses the points raised during the review process.

Three major issues remainFirst, please provide the alternative interpretation of the data expressed by Reviewer 2 or a rebuttal of that viewpoint.Second, the AE question regarding the failure to integrate a single copy of the gene to show complementation was not fully addressed.  What is the evidence that the expected gene has been cloned?Third, while the images are quite good, the pdf is present at limiting resolution.  Be certain that you increase the resolution of the figures and possibly the size of the lane designations so that they will be easily visible in the final version.Fourth, while the authors have addressed most of the issues, the answers to the AE's critique do not specify where in the text (line numbers) specific changes have been made.  In the absence of this information, which by convention should be provided by the authors, the AE is forced to go through multiple re-reads to verify that each point has indeed been addressed. The AE requests that this detail be provided to ensure that every question has been answered.Fifth, please address the additional textual issues raised by Reviewer 2.There are no significant differences between the reviewers.The authors have dealt with each of the issues, although the AE requires more specification of where in the text specific changes have been made. In addition, the issue of the proper identification of the gene that is overproduced must be addressed.

We would appreciate receiving your revised manuscript by Jun 07 2020 11:59PM. To enhance the reproducibility of your results, we recommend that if applicable you deposit your laboratory protocols in protocols.io, where a protocol can be assigned its own identifier (DOI) such that it can be cited independently in the future. For instructions see: http://journals.plos.org/plosone/s/submission-guidelines#loc-laboratory-protocols

We look forward to receiving your revised manuscript.

Kind regards,

Arthur J. Lustig, PhD

Academic Editor

PLOS ONE

Additional Editor Comments (if provided):

Several issues remain that must be addressed.

Reviewers' comments:

Reviewer's Responses to Questions

**Comments to the Author**

1. If the authors have adequately addressed your comments raised in a previous round of review and you feel that this manuscript is now acceptable for publication, you may indicate that here to bypass the “Comments to the Author” section, enter your conflict of interest statement in the “Confidential to Editor” section, and submit your "Accept" recommendation.

Reviewer #1: All comments have been addressed

Reviewer #2: (No Response)

2. Is the manuscript technically sound, and do the data support the conclusions?

Reviewer #1: Yes

Reviewer #2: Yes

3. Has the statistical analysis been performed appropriately and rigorously? 

Reviewer #1: Yes

Reviewer #2: Yes

4. Have the authors made all data underlying the findings in their manuscript fully available?

Reviewer #1: Yes

Reviewer #2: Yes

5. Is the manuscript presented in an intelligible fashion and written in standard English?

Reviewer #1: Yes

Reviewer #2: Yes

6. Review Comments to the Author

Reviewer #1: (No Response)

Reviewer #2: The authors addressed most of the comments. A few minor changes are still suggested.

Minor suggestions:

line 219: "In yeast, galactose-1-phosphate is encoded by the GAL1 gene."

Correct to galactokinase is encoded by the GAL1 gene.

lines 363-365: "The fact that YTA6 and YPR096C compensated the same two gene deletions, further connects their activities together in the context of LiCl sensitivity."

Although this conclusion is possible (and should be kept in the manuscript), another possibility is that the overexpression of YTA6 and YPR096C would improve PGM2 translation, leading to an increase in PGM2 activity in cells that was shown to confer resistance to lithium in galactose medium (Masuda et al., 2001). According to this hypothesis, if the main cause of toxicity under these conditions is the decrease in PGM activity, the overexpression of YTA6 and YPR096C would be "solving" the original problem and thus making any yeast strain more tolerant to lithium, not only those with related function in the cell.

Since at this point we cannot discern between the two hypothesis, I think authors should discuss both possibilities.

line 418: reference # 11.

The citation format for this reference is wrong. Please correct this one, and check carefully for other mistakes in the reference section.

Figures quality: Still the quality of the figures presented in this PDF is low, especially Figure 5 that even if we amplify in a PDF reader, some legends are not readable. Possible that this is a problem of conversion to PDF, but be careful when submitting the final versions of the figure to assure readability in the final version of the manuscript.

7. PLOS authors have the option to publish the peer review history of their article (what does this mean?). If published, this will include your full peer review and any attached files.

Reviewer #1: No

Reviewer #2: Yes: Claudio A Masuda

---

## [Author Response · Author response to Decision Letter 2]

2 Jun 2020

We would like to start by once again thanking Drs. AJ Lusting, CA Masuda and the unanimous reviewer #1 for their invaluable time and comments to improve the quality of this manuscript. 

Editor’s comments:

1- First, please provide the alternative interpretation of the data expressed by Reviewer 2 or a rebuttal of that viewpoint.

Our response: The suggested alternative explanation is now included in the text. Lines 372-377.

2- Second, the AE question regarding the failure to integrate a single copy of the gene to show complementation was not fully addressed. What is the evidence that the expected gene has been cloned?

Our response: Discussion of complementation is now removed from the text and replaced by reporting the phenotypic observations after the introduction of the over-expression plasmid into the corresponding gene deletion mutants. These plasmids were purchased from Thermofisher and their integrity was confirmed by PCR analysis. Lines 81, 82, 195-197, 202 and 203. 

3- Third, while the images are quite good, the pdf is present at limiting resolution. Be certain that you increase the resolution of the figures and possibly the size of the lane designations so that they will be easily visible in the final version.

Our response: Resolution of all figures is now increased.

4- Fourth, while the authors have addressed most of the issues, the answers to the AE's critique do not specify where in the text (line numbers) specific changes have been made. In the absence of this information, which by convention should be provided by the authors, the AE is forced to go through multiple re-reads to verify that each point has indeed been addressed. The AE requests that this detail be provided to ensure that every question has been answered.

Corresponding Associate Editor’s comments from the previous round:

• Specific errors include the definition of NAT (it is not nourseothricin but, rather nourseothricin N-acetyl transferase). 

Our response: NAT is now replaced by clonNAT (nourseothricin sulfate). Line 89.

• How was the concentration of LiCl determined? What were the effects in different concentration of LiCl> and a clear relationship between context for which methods are involved in which experiments.

Our response: Proper references are now included for the starting concentrations of LiCl. Additional explanation is also provided. Description is added for each method connecting it to the corresponding experiment. Lines 114-116, 170-171 and 227-228.

• Is the statistical basis of the qPCR the MIQE convention? If not, please redo these evaluations according the this convention.

Our response: Yes, it is. The text is now modified to reflect this. Line 137.

• Define SDL.

Our response: The text is now modified accordingly. SDL is removed from line 165 as this form of genetic interaction is not included in the manuscript.

• Statistical tests should include the sets of values that are being compared (for example in Western blot analysis). Also place n value and statistical test in Figure Legends.

Our response: The text is now modified accordingly. Lines 211, 215, 217, 219, 256-258, 276-280, 301-305, 354-355, and 387-388. 

• Please define the unconventional use of "fitness". Are you not referring to colony size?

Our response: A description is now added in the Materials and methods section. In Genetic Interaction analysis, colony size is commonly used as a measure of fitness. For example: Tong et al 2001 Science; Toufighi et al 2011 Nature Methods; Roguev et al 2018 Cold Spring Harb Protoc; etc. Lines 162-163.

• Please describe the tests for the functionality of fusion proteins used in this study. Also clarify which experiments used and did not use the fusion protein.

Our response: PCR analysis and LiCl sensitivity was used to confirm the integrity of Pgm2p-GFP strain. The Material and methods section is modified accordingly. Lines 82-85.

• Much of the data is presented graphically in Figures 2,3,4,5. An example of each dataset is required in these Figures. Similarly, a subset of the data interactions summarized in Figure 6 that have been uncovered in this paper should be presented. In addition, in a Supplementary Data section, by PLOS One regulations, all of the raw data must be presented so that the data is can be directly assessed.

Our response: Supporting information for each data set is added. An example of each dataset is added to each figure in lines 215, 277, 302 and 355. Table S1, S2 and S3 in lines 544-553 are added as supporting information data and are mentioned in the manuscript in lines 322, 325, 326, and 330. Figure 6, a representative interaction is now included; also see line 355. 

• The logic behind the interpretation of mutations in the presence of gal1 null alleles must be explained more clearly as noted by Reviewer 2.

Our response: The text is now modified accordingly. Lines 221-222.

• The isolation of interacting genes in some ribosomal protein genes and regulators of translation implicates the translation process. However, conclusions the basis of these studies should be in line with the strength of the data.

Our response: The text is now modified to reduce the strength of the concluding sentence regarding GI data. Lines 347-349. 

Reviewer #2:

1- line 219: "In yeast, galactose-1-phosphate is encoded by the GAL1 gene."Correct to galactokinase is encoded by the GAL1 gene.

Our response: The text is now modified accordingly. Line 222. 

2- lines 363-365: "The fact that YTA6 and YPR096C compensated the same two gene deletions, further connects their activities together in the context of LiCl sensitivity."

Although this conclusion is possible (and should be kept in the manuscript), another possibility is that the overexpression of YTA6 and YPR096C would improve PGM2 translation, leading to an increase in PGM2 activity in cells that was shown to confer resistance to lithium in galactose medium (Masuda et al., 2001). According to this hypothesis, if the main cause of toxicity

under these conditions is the decrease in PGM activity, the overexpression of YTA6 and YPR096C would be "solving" the original problem and thus making any yeast strain more tolerant to lithium, not only those with related function in the cell.

Since at this point we cannot discern between the two hypothesis, I think authors should discuss both possibilities.

Our response: The text is now modified accordingly. Added to manuscript from lines 369-375.

3- line 418: reference # 11.The citation format for this reference is wrong. Please correct this one, and check carefully for other mistakes in the reference.

Our response: The text is now modified accordingly. 

4- Figures quality: Still the quality of the figures presented in this PDF is low, especially Figure 5 that even if we amplify in a PDF reader, some legends are not readable. Possible that this is a problem of conversion to PDF, but be careful when submittingthe final versions of the figure to assure readability in the final version of the manuscript.

Our response: Resolution of all figures is now increased.

---

## [Decision Letter · Decision Letter 3]

9 Jun 2020

Sensitivity of yeast to lithium chloride connects the activity of YTA6 and YPR096C to translation of structured mRNAs

PONE-D-19-21480R3

Dear Dr. Golshani,

We’re pleased to inform you that your manuscript has been judged scientifically suitable for publication and will be formally accepted for publication once it meets all outstanding technical requirements.

Kind regards,

Arthur J. Lustig, PhD

Academic Editor

PLOS ONE

Additional Editor Comments (optional):

Reviewers' comments:

Reviewer's Responses to Questions

**Comments to the Author**

1. If the authors have adequately addressed your comments raised in a previous round of review and you feel that this manuscript is now acceptable for publication, you may indicate that here to bypass the “Comments to the Author” section, enter your conflict of interest statement in the “Confidential to Editor” section, and submit your "Accept" recommendation.

Reviewer #2: All comments have been addressed

2. Is the manuscript technically sound, and do the data support the conclusions?

Reviewer #2: (No Response)

3. Has the statistical analysis been performed appropriately and rigorously? 

Reviewer #2: (No Response)

4. Have the authors made all data underlying the findings in their manuscript fully available?

Reviewer #2: (No Response)

5. Is the manuscript presented in an intelligible fashion and written in standard English?

Reviewer #2: (No Response)

6. Review Comments to the Author

Reviewer #2: (No Response)

7. PLOS authors have the option to publish the peer review history of their article (what does this mean?). If published, this will include your full peer review and any attached files.

Reviewer #2: Yes: Claudio A Masuda

---

## [Editor Report · Acceptance letter]

12 Jun 2020

PONE-D-19-21480R3 

Sensitivity of yeast to lithium chloride connects the activity of *YTA6* and *YPR096C* to translation of structured mRNAs 

Dear Dr. Golshani:

I'm pleased to inform you that your manuscript has been deemed suitable for publication in PLOS ONE. Congratulations! Your manuscript is now with our production department. 

Kind regards, 

on behalf of

Dr. Arthur J. Lustig 

Academic Editor

PLOS ONE